https://doi.org/10.1038/s42003-023-05311-1　　**OPEN**
# The citrate transporters SLC13A5 and SLC25A1 elicit different metabolic responses and phenotypes in the mouse

Gonzalo Fernandez-Fuente [1,2], Katherine A. Overmyer[3,4], Alexis J. Lawton [3,5], Ildiko Kasza[6], Samantha L. Shapiro[1,2], Patricia Gallego-Muñoz[7], Joshua J. Coon [3,4,8], John M. Denu[3,5], Caroline M. Alexander[6] & Luigi Puglielli [1,2,9,10 ✉]

Cytosolic citrate is imported from the mitochondria by SLC25A1, and from the extracellular milieu by SLC13A5. In the cytosol, citrate is used by ACLY to generate acetyl-CoA, which can then be exported to the endoplasmic reticulum (ER) by SLC33A1. Here, we report the generation of mice with systemic overexpression (sTg) of SLC25A1 or SLC13A5. Both animals displayed increased cytosolic levels of citrate and acetyl-CoA; however, SLC13A5 sTg mice developed a progeria-like phenotype with premature death, while SLC25A1 sTg mice did not. Analysis of the metabolic profile revealed widespread differences. Furthermore, SLC13A5 sTg mice displayed increased engagement of the ER acetylation machinery through SLC33A1, while SLC25A1 sTg mice did not. In conclusion, our findings point to different biological responses to SLC13A5- or SLC25A1-mediated import of citrate and suggest that the directionality of the citrate/acetyl-CoA pathway can transduce different signals.

[1] Department of Medicine, University of Wisconsin-Madison, Madison, WI, USA. [2] Waisman Center, University of Wisconsin-Madison, Madison, WI, USA. [3] Department of Biomolecular Chemistry, University of Wisconsin-Madison, Madison, WI, USA. [4] Morgridge Institute for Research, Madison, WI, USA. [5] Wisconsin Institute for Discovery, University of Wisconsin-Madison, Madison, WI, USA. [6] McArdle Laboratory for Cancer Research, University of Wisconsin-Madison, Madison, WI, USA. [7] Department of Cell Biology, Genetics, Histology and Pharmacology, Faculty of Medicine, University of Valladolid, Valladolid, Spain. [8] Department of Chemistry, University of Wisconsin-Madison, Madison, WI, USA. [9] Geriatric Research Education Clinical Center, Veterans Affairs Medical Center, Madison, WI, USA. [10] Department of Neuroscience, University of Wisconsin-Madison, Madison, WI, USA. ✉email: lp1@medicine.wisc.edu

Cells have developed nutrient/metabolite sensing mechanisms to respond to environmental fluctuations occurring either intracellularly or extracellularly. These mechanisms allow them to maintain homeostasis and ensure crosstalk between different organelles and compartments[1–3]. Therefore, key cellular metabolites that reflect the immediate activity of specific metabolic enzymes and/or the functional metabolic state of intracellular organelles have emerged as powerful signaling regulators. The mechanisms through which they exert their functions and influence cellular regulatory events are very complex and still largely unknown. How these mechanisms act at the organismal level and influence systemic phenotypes/diseases is even more complex and less known.

The citrate/acetyl-CoA pathway is a recognized key node in metabolism that has emerged as a novel branch of the more general nutrient-signaling pathway[2–5]. Compelling evidence indicates that protein acetylation across multiple organelles and compartments reflects the metabolic status of the cell. Evidence also indicates that many proteins and biochemical pathways are directly modulated by Nε-lysine acetylation through the availability and the intracellular compartmentalization of acetyl-CoA, donor of the acetyl group for Nε-lysine acetylation[1,3,5].

Cytosolic acetyl-CoA is mainly generated by ATP Citrate Lyase (ACLY), which uses citrate and CoA[6]. Citrate is obtained through active transport from the mitochondria by the citrate/malate antiporter SLC25A1/CTP, and from the plasma membrane by the citrate/Na$^+$ symporter SLC13A5/NACT[3]. Cytosolic acetyl-CoA has free access to the nucleus and can also serve as a substrate for nuclear-based protein acetylation. Active transport of acetyl-CoA from the cytosol into the lumen of the endoplasmic reticulum (ER) is ensured by the ER membrane transporter SLC33A1/AT-1, which exchanges acetyl-CoA for free CoA in an antiporter fashion[6]. Therefore, the interplay between different membrane transporters ultimately influences organelle-specific adaptive responses to a specific metabolic signal: the intracellular availability and flux of citrate and acetyl-CoA (see Fig. 1a).

Mutations, microdeletions, and gene duplication events within SLC25A1/CIC, SLC13A5/NACT and SLC33A1/AT-1 have been associated with different diseases spanning from developmental delay with multi-system deficits to progeria-like features, autism spectrum disorder (ASD) and attention deficit hyperactivity disorder, different forms of epileptic encephalopathy, as well as peripheral forms of neuropathy[3,7–14]. Thus, genetic alterations affecting the three membrane transporters that are key to the intracellular flux and availability of citrate and acetyl-CoA are linked to severe human diseases.

In this study, we report the generation and phenotypic characterization of mice with systemic overexpression of either SLC25A1 or SLC13A5. We show that the animals develop strikingly different phenotypes and completely different metabolic signatures within the lipidome and the acetylproteome, indicating that the cell is able to differentiate between SLC25A1- and SLC13A5-dependent pools of citrate and acetyl-CoA. We also show that SLC13A5 is a much more powerful regulator of AT-1 activity and ER acetylation than SLC25A1, suggesting that the directionality of the citrate/acetyl-CoA pathway can transduce different signals. As a result, the lethal phenotype displayed by SLC13A5 mice can be rescued by inhibiting the two ER luminal-based acetyltransferases, ATase1 and ATase2, which act downstream of AT-1 to acetylate ER cargo proteins.

## Results

### SLC13A5 sTg, but not SLC25A1 sTg, mice display a progeria-like phenotype. To study the systemic role of citrate transporters SLC25A1 and SLC13A5, we generated transgenic (sTg) mice with an inducible overexpression Tet-Off system driven by the Rosa26 locus (Fig. 1a, b). For the purpose of this study, the animals with systemic overexpression of SLC25A1 (referred to as SLC25A1 sTg) were maintained in the absence of doxycycline; therefore, they overexpressed SLC25A1 since conception and throughout their entire life, including development. The animals were born with Mendelian ratio and were completely normal at birth; however, within 1 month they appeared smaller than their wild-type (WT) littermates (Fig. 1c). We did not observe any effect on the lifespan of the animals, nor any phenotypic manifestation resembling progeria (Table 1). On the contrary, animals with systemic overexpression of SLC13A5 from conception (referred to as SLC13A5 sTg$^{OC}$) were born with Mendelian ratio but died immediately after birth. Therefore, we also generated mice where the overexpression of SLC13A5 was induced at birth (referred to as SLC13A5 sTg$^{OB}$) or weaning (referred to as SLC13A5 sTg$^{OW}$) (see Fig. 1d).

At visual inspection, recently born SLC13A5 sTg$^{OC}$ displayed multiple skin lesions, suggesting that death was caused by the mechanical stress associated with transiting through the birth canal (Fig. 1e). When analyzed at embryonic day 18 (E18), the skin of SLC13A5 sTg$^{OC}$ mice exhibited a narrow stratum granulosum with coarsely clumped keratohyaline granules (Supplementary Fig. 1a). Importantly, the differentiation of embryonic skin (E18) was severely impacted by SLC13A5 over-expression. The basal (keratin 5; K5-positive) layer was hyperplastic, increasing in thickness by 2-fold, and becoming 3 cells thick instead of a mostly single basal layer. Consistent with this observation, almost 50% of K5-positive basal cells were in mitosis (Supplementary Fig. 1b, c). The thickness of the more differentiated cell layers (spinous and above) was relatively unaffected by SLC13A5 over-expression (assessed by keratin 10 (K10) staining; Supplementary Fig. 1b). However, judged from the histology, the differentiation of suprabasal keratinocytes was inhibited, revealed as a depletion of the granular layer, decreased cytoplasmic volume in suprabasal keratinocytes, and a depleted enucleated cornified layer. A preliminary attempt to test the skin barrier function showed no difference (Supplementary Fig. 1d). Perhaps most surprising was the massive increase in dermal cell mitotic index (Supplementary Fig. 1c) in SLC13A5 sTg$^{OC}$ skins, together with a 2-fold thickening of dermis. This would suggest that the dermis may be the origin of skin dysfunction, as it is for several diseases and wound healing of mouse and human skins[15].

SLC13A5 sTg$^{OB}$ and SLC13A5 sTg$^{OW}$ were born with Mendelian ratio and were completely normal at birth. However, after approx. 2 and 10 months, respectively, they displayed a severe phenotype (Fig. 1d, f; Table 1; Supplementary Movie 1) that was reminiscent of segmental forms of progerias[16–19]. SLC13A5 sTg$^{OB}$ mice remained smaller throughout their entire life as compared to WT littermates (Fig. 1g) or even SLC25A1 sTg (Fig. 1h). Both SLC13A5 sTg$^{OB}$ and SLC13A5 sTg$^{OW}$ displayed a very short lifespan (Fig. 1i). The skin displayed hair loss, disseminated lesions and reduced wound repair, and most of the animals developed a hunched posture and rectal prolapse (Table 1; Supplementary Movie 1). Evidence of systemic overexpression of the transporters was documented by Western blot (Fig. 1j, k). SLC13A5 sTg$^{OB}$ mice (simply referred to as SLC13A5 sTg thereafter) were used as the main model for our study.

### SLC13A5 sTg mice display systemic inflammation, skin and cornea alterations, reduced bone density and cellular senescence. Histological assessment of SLC13A5 sTg mice revealed aberrant skin structure with hyper-proliferative features of the different layers, aberrant keratin accumulation, and inflammatory

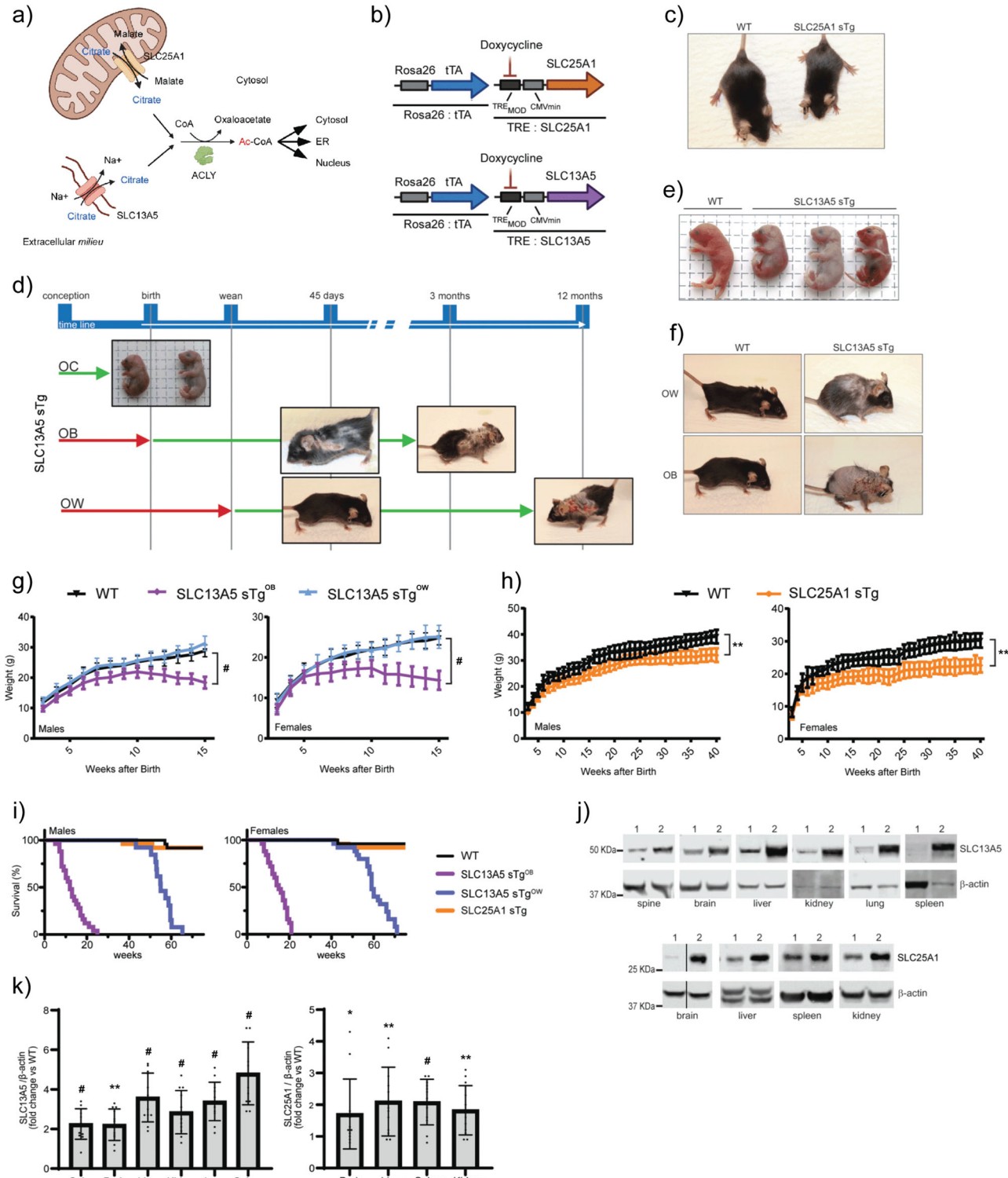

**Fig. 1 SLC13A5 sTg and SLC25A1 sTg display different phenotypes. a** Schematic view of the citrate/acetyl-CoA pathway. Image was generated using BioRender. **b** SLC13A5 sTg and SLC25A1 sTg mice were generated with an inducible Tet-Off expression system under the control of the Rosa26 locus for systemic overexpression. **c** SLC25A1 sTg mouse and WT littermate when 75 days old. **d** SLC13A5 sTg mice were studied under three different experimental conditions based on the starting point of overexpression: overexpression from conception (OC), overexpression from birth (OB), and overexpression from weaning (OW). **e** SLC13A5 sTg^OC mice and WT littermate at birth. **f** Representative SLC13A5 sTg^OW, SLC13A5 sTg^OB, and WT littermates when 125 days old. **g** Body weight of male and female WT, SLC13A5 sTg^OB, and SLC13A5 sTg^OW ($n = 20$/group). $^\#P < 0.0005$ via mean comparison using Student's $t$ test. **h** Body weight of male and female WT and SLC25A1 sTg animals ($n = 20$/group). $^{**}P < 0.005$ via mean comparison using Student's $t$ test. **i** Lifespan of WT, SLC25A1 sTg, SLC13A5 sTg^OW and SLC13A5 sTg^OB mice ($n = 25$/group). Maximum lifespan of SLC13A5 sTg^OB mice: males = 175 days, females = 147 days; $P < 0.0005$ via Kaplan–Meier lifespan test. **j** Representative Western blots showing SLC13A5 and SLC25A1 overexpression in different tissues (1, WT mice; 2, sTg mice). **k** Total protein quantification of different tissues from SLC13A5 and SLC25A1 sTg mice displayed as fold of change vs WT animals ($n = 12$/group). $^*P < 0.05$, $^{**}P < 0.005$, $^\#P < 0.0005$ via mean comparison using Student's $t$ test.

| Table 1 Observed phenotype of SLC13A5 and SLC25A1 sTg mice. | | |
|---|---|---|
| | SLC13A5[OB] | SLC25A1 |
| Median lifespan | Reduced | Normal |
| Maximum lifespan | Reduced | Normal |
| Body weight | Reduced | Reduced |
| Hunched posture | Common | Normal |
| Hair loss | Pronounced | Normal |
| Hair regrowth | Reduced | Normal |
| Skin lesions | Pronounced | Normal |
| Wound repair | Retarded | Normal |
| Dermal thickness | Increased | Normal |
| Adipose tissue | Reduced | Reduced |
| Bone density | Reduced | Normal |
| Liver size | Normal | Reduced |
| Spleen size | Increased | Normal |
| Lymph nodes size | Increased | Normal |
| Gallbladder size | Normal | Enlarged |
| Rectal prolapse | Common | Normal |
| Systemic inflammation | Increased | Normal |
| Peripheral WBC | Neutrophilia | Normal |
| Peripheral RBC | Normal | Normal |
| Citrate (cytoplasm) | Increased | Increased |
| Ac-CoA (cytoplasm) | Increased | Increased |
| Ac-CoA (ER) | Increased | Increased |
| Glycemia (fasting) | Normal | Normal |
| Ketone bodies (fasting) | Increased | Increased |
| Lipid droplets (number) | Increased | Increased |
| Lysosome (number) | Increased | Normal |
| Mitochondria (surface) | Increased | Increased |

*WBC white blood cell count, RBC red blood cell count*

infiltration (Fig. 2a). Both males and females displayed severe bone density loss, which was reminiscent of osteoporosis (Fig. 2b). Postmortem examination revealed splenomegaly (Fig. 2c), enlarged lymph nodes (Fig. 2d), and tissue inflammation (Fig. 2e, f). Chronic tissue inflammation is often associated with markers of cellular senescence[20–23]. Therefore, we analyzed levels of *p16*, *p21* and β-galactosidase (SA-β-gal), three established markers of cell senescence. We consistently found increased levels of all three markers in the SLC13A5 sTg mice (Fig. 2g, h).

About 10–15% of SLC13A5 sTg mice showed severe corneal changes, which included exaggerated epithelia thickening, abnormal epithelial stratification with keratinization, loss of stromal collagen regular alignment with cellular infiltration, and neovascularization. All these alterations manifested with severe loss of corneal transparency in the living animal (Fig. 2i).

**SLC25A1 and SLC13A5 sTg mice have altered intracellular compartmentalization of citrate and acetyl-CoA, steatosis and ketosis**. Both SLC25A1 and SLC13A5 transport citrate into the cytoplasm and feed into the citrate/acetyl-CoA pathway through ACLY-mediated conversion of citrate and free CoA into acetyl-CoA. Cytosolic acetyl-CoA can then be transported into the ER by the ER membrane transporter AT-1 where it is utilized by two ER-based acetyltransferases (Fig. 3a). To analyze the metabolic profile induced by the overexpression of the individual transporters, we performed subcellular fractionation of the liver and resolved cell lysates, cytosol, ER, and mitochondria. As expected, the cytosolic fraction of both SLC25A1 sTg and SLC13A5 sTg mice displayed higher levels of citrate and acetyl-CoA when compared to WT littermates (Fig. 3b, c). The increased cytosolic levels of acetyl-CoA also translated into increased import of acetyl-CoA in the ER lumen (Fig. 3b). These data show that overexpression of either SLC25A1 or SLC13A5 altered the

intracellular compartmentalization of citrate and acetyl-CoA, resulting in increased steady-state levels of acetyl-CoA in both the cytosol and ER lumen. SLC25A1 acts as an antiporter with concurrent transport of citrate to the cytoplasm and malate to the mitochondria (Fig. 3a). Consistently, SLC25A1 sTg mice displayed increased levels of malate in the mitochondria with corresponding reduced levels in the cytosol (Fig. 3d). As expected, SLC13A5 sTg mice did not show altered compartmentalization of malate (Fig. 3d).

Histological assessment revealed that both SLC25A1 sTg and SLC13A5 sTg mice displayed increased propensity to liver steatosis, which could be documented by both LipidTOX staining and electron microscopy (Fig. 3e, f). In the case of the SLC25A1 sTg mice, lipid droplets were not only more abundant, but were also larger than those of WT and SLC13A5 sTg mice. SLC25A1 sTg livers were also smaller in size when compared to WT littermates (Fig. 3g). Metabolic assessment of SLC25A1 sTg and SLC13A5 sTg mice revealed normal levels of circulating glucose but increased levels of beta-hydroxybutyrate, indicative of ketosis and increased β-oxidation of fatty acids (Fig. 3h).

**Mice with systemic overexpression of SLC13A5 and SLC25A1 display different Nε-lysine acetylation profiles**. Acetyl-CoA dependent processes include lipid metabolism, citric acid cycle, and Nε-lysine acetylation, which can occur in the nucleus and cytoplasm[24], ER[3,25], mitochondria[26], and peroxisomes[27]. Protein acetylation is a mechanism of post-translational regulation that can affect protein interactions and function, and can be independent of protein expression changes[28]. Therefore, it is possible that fluctuations in acetyl-CoA availability occurring within the range of affinity constants of different Nε-lysine acetyltransferases might affect the acetylation status and the function of different target proteins within different cellular organelles and compartments.

To determine whether that was the case, and to begin dissecting the underlying phenotypic differences of the SLC13A5 and SLC25A1 sTg models at the mechanistic level, we performed quantitative acetyl-proteomics using the liver as target tissue. Specifically, we quantified steady-state acetylation stoichiometry on lysine sites at the proteome level. In order to increase detection resolution, we resolved cytoplasmic, nuclear/membrane, and chromatin-associated fractions before measuring the stoichiometry of acetylation[29,30]. In the three fractions combined, we detected a total of 6677 lysine sites in WT (2049 proteins), 6534 lysine sites in SLC13A5 (2014 proteins), and 6381 lysine sites in SLC25A1 sTg mice (1988 proteins), respectively. When compared to WT, we found 727 lysine sites (599 proteins) in SLC13A5 sTg mice and 733 lysine sites (591 proteins) in SLC25A1 sTg mice that had a statistically different acetylation stoichiometry (Fig. 4a). When we compared the statistically significant stoichiometry changes of SLC13A5 sTg and SLC25A1 sTg models, we observed remarkable differences among the two sTg models with only approx. 30% (224 sites and 193 proteins) overlap (Fig. 4a–c).

Kyoto Encyclopedia of Genes and Genomes (KEGG) pathway analysis of significantly changed acetylated proteins revealed similar as well as distinct categories between SLC13A5 sTg and SLC25A1 sTg mice (Fig. 4d–f). Common pathways were pertinent to protein biosynthesis/processing, fatty acid metabolism/degradation, and peroxisome-based acetyl-CoA/acyl-CoA metabolism (Fig. 4d), while divergent pathways were pertinent to different metabolic processes (Fig. 4e, f).

In addition, for both models, we evaluated the subcellular location of all acetylated proteins that were significantly different from WT. The general trend indicates increased acetylation in both models (Fig. 4g). The 224 acetylation sites shared by the two

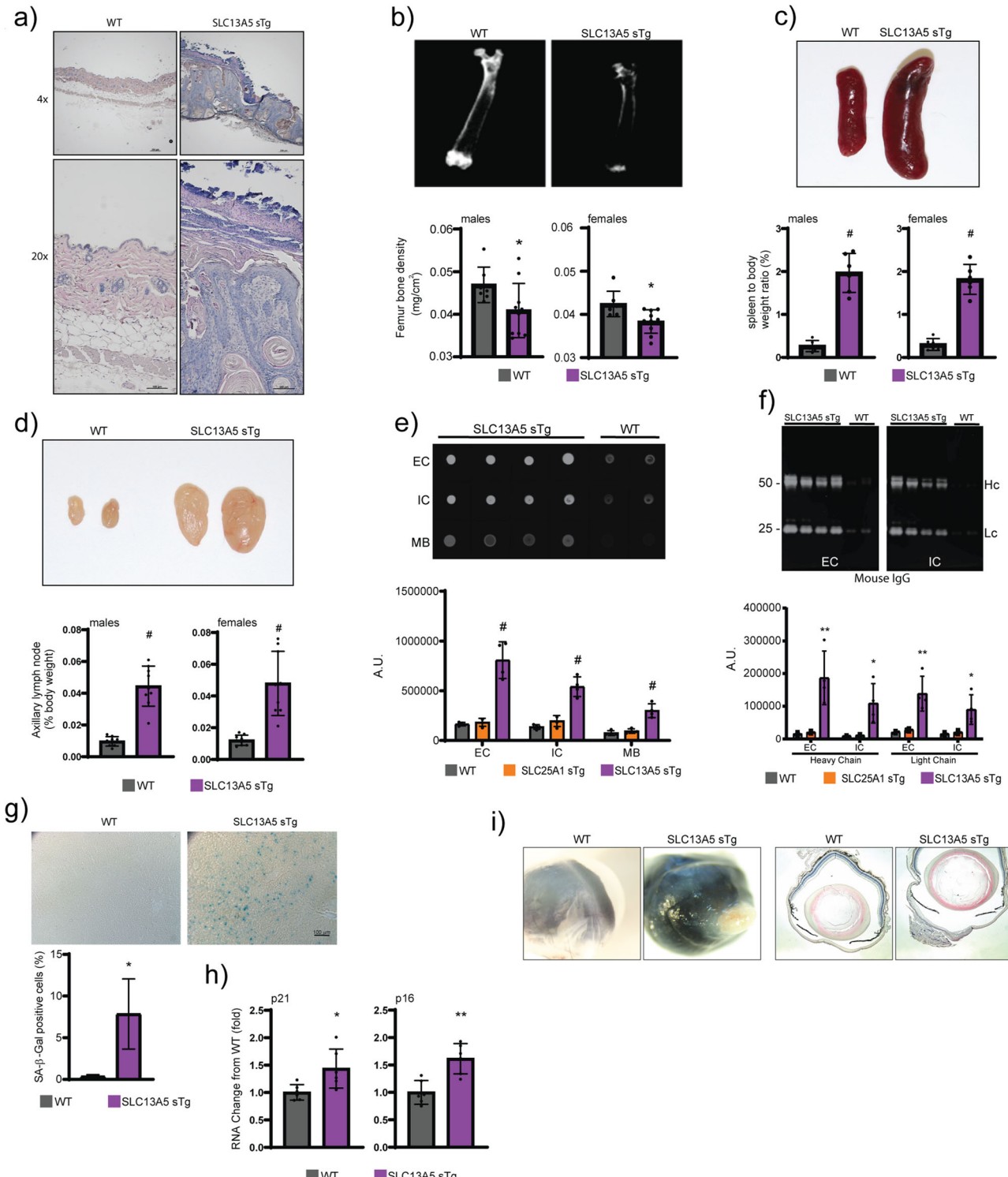

**Fig. 2 SLC13A5 sTg mice display a systemic phenotype that is consistent with a segmental form of progeria. a** Representative H&E staining of skin sections from WT and SLC13A5 sTg mice. **b** Faxitron X-ray (femur) and bone mineral density quantification of WT and SLC13A5 sTg mice (WT, $n = 7$; SLC13A5 sTg, $n = 10$). *$P < 0.05$ via mean comparison using Student's $t$ test. **c** Image and weight of whole spleen from WT and SLC13A5 sTg mice ($n = 6$/group). #$P < 0.0005$ via mean comparison using Student's $t$ test. **d** Image and weight of axillary lymph nodes from WT and SLC13A5 sTg mice ($n = 9$/group). #$P < 0.0005$ via mean comparison using Student's $t$ test. **e** Dot blot of liver tissue immunoglobulins determined with anti-mouse IgG. Representative images and quantification of results (WT, $n = 4$; SLC13A5 sTg, $n = 4$). #$P < 0.0005$, via mean comparison using Student's $t$ test. Extracellular, intracellular, and membrane fractions (EC, IC, MB). **f** Western blot showing liver tissue immunoglobulins determined with anti-mouse IgG and quantification of results (WT, $n = 4$; SLC13A5 sTg, $n = 4$). *$P < 0.05$, **$P < 0.005$ via mean comparison using Student's $t$ test. Extracellular and intracellular fractions (EC, IC). Heavy chain and light chain (Hc, Lc). **g** SA-β-Gal staining of liver slides from WT and SLC13A5 sTg mice. Representative images and quantification of results ($n = 4$ mice/group; 150 cells/animal). *$P < 0.05$ via mean comparison using Student's $t$ test. **h** p21 and p16 mRNA quantification in liver ($n = 6$/group). **$P < 0.005$ via mean comparison using Student's $t$ test. **i** Representative images and H&E stained cross-sections of the cornea from WT and SLC13A5 sTg mice.

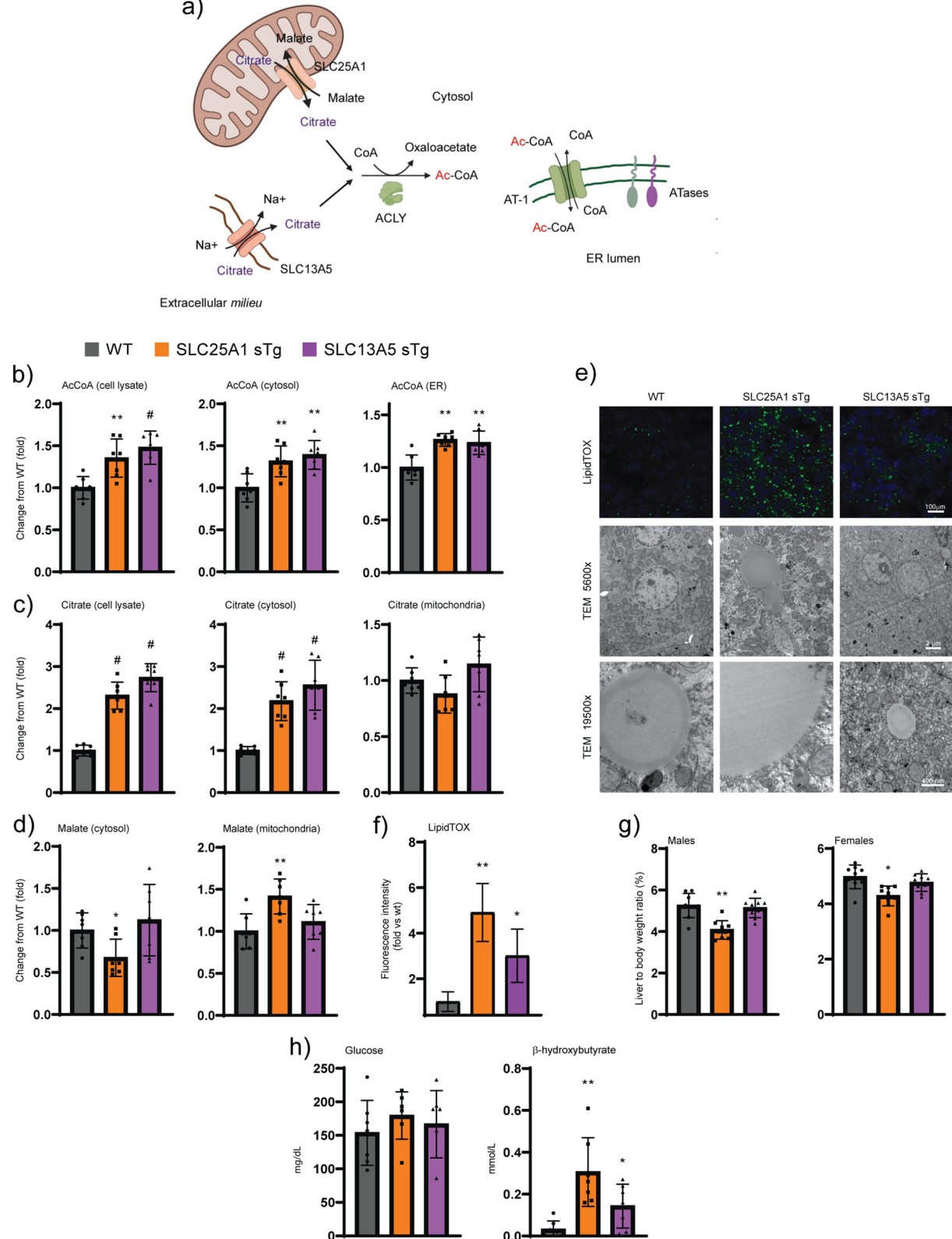

sTg models that displayed statistically significant divergence from WT mice were distributed as follows: 150 in the cytosol, 35 in the mitochondria, 22 in the nucleus, and 17 in the secretory pathway (Supplementary Fig. 2). The gene-network plot constructed with the 193 proteins that are shared among the two models using the GO cellular component function database revealed categories related to the protein biosynthetic (translational) machinery, as well as metabolic pathways associated with the mitochondria and the peroxisomes (Fig. 4h). A similar output was observed when building STRING-based networks (Supplementary Fig. 3a).

When SLC13A5 sTg mice were compared to SLC25A1 sTg mice, we found 698 acetylation sites distributed on 500 proteins being uniquely affected by the overexpression of SLC13A5 (Fig. 5a, b). KEGG pathways that were highly represented in

**Fig. 3 SLC25A1 sTg and SLC13A5 sTg mice display elevated citrate and acetyl-CoA levels, lipid accumulation, and ketosis. a** Schematic view of potential metabolic effects triggered by the overexpression of SLC25A1 and SLC13A5. Image was generated using BioRender. **b** Total (cell lysate), cytosolic, and ER levels of acetyl-CoA expressed as arbitrary units ($n = 7$/group). *$P < 0.05$, **$P < 0.005$, #$P < 0.0005$ via mean comparison using Student's $t$ test. **c** Total (cell lysate), cytosolic, and mitochondria levels of citrate expressed as arbitrary units ($n = 7$/group). #$P < 0.0005$ via mean comparison using Student's $t$ test. **d** Cytosolic and mitochondria levels of malate expressed as arbitrary units ($n = 7$/group). *$P < 0.05$ and **$P < 0.005$ via mean comparison using Student's $t$ test. **e** Representative LipidTOX green staining and electron microscopy of liver sections. **f** Quantification of the fluorescence intensity from LipidTOX green staining expressed as arbitrary units ($n = 10$/group; 50 cells/animal). *$P < 0.05$ and **$P < 0.005$ via mean comparison using Student's $t$ test. **g** Liver to body weight ratio ($n = 9$/group). *$P < 0.05$ and **$P < 0.005$ via mean comparison using Student's $t$ test. **h** Fasting levels of blood glucose and β-hydroxybutyrate ($n = 7$/group). *$P < 0.05$ and **$P < 0.005$ via mean comparison using Student's $t$ test.

---

the SLC13A5 model included processes related to protein biosynthesis, fatty acid metabolism, detoxification of xenobiotics, and drug metabolism (Fig. 5c). The differential response of the SLC13A5 manipulation was observed across different organelles and compartments (Fig. 5d; Supplementary Fig. 4). The gene-network plot representing the same 500 proteins using the GO cellular component function database revealed a large representation of the protein biosynthetic machinery and ER-dependent translation. Also represented were proteins involved with metabolic processes within different cellular compartments (Fig. 5e). A similar profile was observed when STRING-based networks were constructed (Supplementary Fig. 3b).

When taken together, these data indicate very different adaptive responses across the lysine acetylation landscape within multiple cellular compartments and organelles. These differences were observed regardless of the inclusion of the WT counterpart in the data analysis process. In essence, although both transporters provide citrate to the cytosol, the cellular response triggered by the increased influx of citrate was very distinct.

**Mice with systemic overexpression of SLC13A5 and SLC25A1 display different lipid profiles.** To complement the acetyl-proteomic data, we performed quantitative lipidomic analysis of liver, serum, skin, and brain of WT, SLC13A5 sTg and SLC25A1 sTg mice. Different lipid species, as well as classes of lipids, were found to be statistically different from WT mice across our sTg models (Fig. 6; Supplementary Figs. 5–7).

In the liver, a total number of 492 lipids were annotated with molecular species or species level identifications[31]. When compared to WT, 68 and 21 lipid species were significantly different in SLC13A5 and SLC25A1 sTg mice, respectively (Fig. 6a). Comparison of the two mouse models revealed a completely different lipid adaptive response to the increased influx of citrate. Indeed, only one species, PC 34:6, was in common (Fig. 6a). Furthermore, we observed a trend toward increased lipid species levels in the SLC13A5 and reduced lipid specie levels in the SLC25A1 model (Fig. 6b, c). When analyzed as lipid groups, the serine (S), phosphatidylserine (PS), and S/PS-related classes emerged with the highest level of upregulation within SLC13A5 sTg mice (Fig. 6d).

The lipid adaptive response in the two sTg models varied among the different tissues analyzed. The serum and the skin mirrored the liver profile with more identified lipids that reached statistical significance in SLC13A5 sTg mice (Supplementary Figs. 5–6), while the brain showed more identified lipids in SLC25A1 sTg mice (Supplementary Fig. 7). However, the overall trend as well as individual lipid species and classes of lipids diverged among the two models across all tissues analyzed.

When taken together, the above data indicate that, although they both supply citrate to the cytosol, SLC13A5 and SLC25A1 elicit very different metabolic responses, which likely reflect the striking phenotypic differences observed. Importantly, the lipid adaptive response appeared to mirror the stoichiometry of acetylation data.

**Hyperactivity of the ER acetylation machinery contributes to the progeria-like phenotype of SLC13A5 sTg mice.** Cytosolic acetyl-CoA is imported into the ER lumen by AT-1 through an antiporter mechanism that involves exchange of cytosolic acetyl-CoA for ER-luminal free CoA (Fig. 7a)[3]. The antiporter mechanism allows AT-1 to respond to changes in availability of acetyl-CoA and free CoA across the ER membrane[3,32–34]. Mice with systemic overexpression of AT-1 (AT-1 sTg) develop a segmental form of progeria that can be rescued by inhibition of the ER-based acetyltransferases, ATase1 and ATase2[32,35]. In light of the phenotypic similarities between AT-1 sTg and SLC13A5 sTg mice, we explored the possibility that the phenotype of SLC13A5 sTg was, at least in part, caused by increased acetylation of ER cargo proteins.

Immunoblot assessment revealed a 3-fold upregulation of endogenous AT-1 in SLC13A5 sTg mice, reaching levels of expression that were almost similar to those observed in AT-1 sTg mice (Fig. 7b, c). AT-1 upregulation was not observed in SLC25A1 sTg mice (Fig. 7b). A major difference between SLC13A5 and SLC25A1 is that the former is a high-capacity citrate transporter, while the latter is a low-capacity transporter (discussed later). Therefore, we can speculate that the increased levels of endogenous AT-1 in the SLC13A5 model reflects an attempt to compensate for the increased availability of acetyl-CoA in the cytosol. In other words, the antiporter mechanism is sufficient in SLC25A1 sTg mice, but not in SLC13A5 sTg mice. Although we observed similar steady-state levels of acetyl-CoA in the ER of the two sTg models (Fig. 3b), it is possible that the excess acetyl-CoA in the SLC13A5 model is being used to acetylate ER cargo proteins and that, therefore, the steady-state levels do not truly reflect the underlying biology (see also Fig. 7j). To verify this hypothesis, we isolated the ER from both sTg models and compared levels of Nε-lysine acetylation of ER cargo proteins. The results show increased acetylation in both SLC25A1 and SLC13A5 sTg mice, as compared to WT mice. However, the levels of acetylation in the SLC13A5 model were much higher than in the SLC25A1 model and appeared comparable to those observed in AT-1 sTg mice (Fig. 7d). SLC13A5 sTg mice also displayed increased Nε-lysine acetylation of nuclear-associated proteins (Supplementary Fig. 8a, b) but normal levels of global acetylation within the total lysate (Supplementary Fig. 8c). Interestingly, the increased acetylation of the nuclear compartment was not as dramatic as the ER compartment (compare Fig. 7d to Supplementary Fig. 8a, b). Overall, these results point, again, to a selective adaptive response and appear to reflect the acetyl-proteome data (Figs. 4 and 5). The progeria phenotype of AT-1 sTg mice is mechanistically linked to the increased acetylation of ER-bound ATG9A, resulting in defective proteo-static control[32,35]. Direct assessment of the acetylation status of ER-bound ATG9A also demonstrated hyper-acetylation of ATG9A in SLC13A5 sTg mice (Fig. 7e). Importantly, the hyper-acetylation of ATG9A (approx. 4-fold; Fig. 4e) appeared to match the upregulation of AT-1 (approx. 3-fold; Fig. 4b). Next, we treated SLC13A5 sTg mice with an oral formulation of

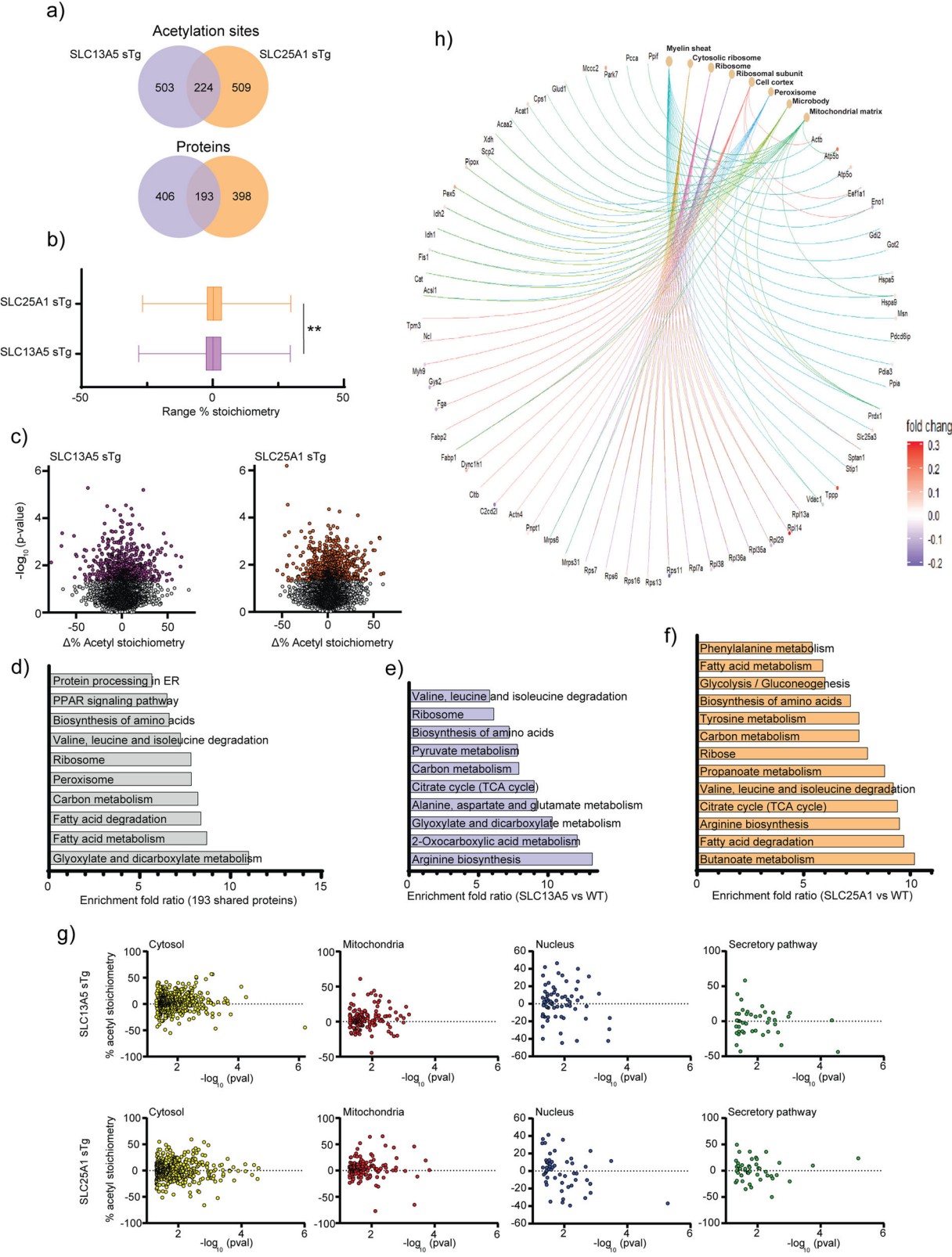

compound 9 (C9), a specific inhibitor of the ATases (see Fig. 7a)[32,36,37]. C9 successfully reduced the levels of Nε-lysine acetylation of ER cargo proteins (Fig. 7d), the acetylation status of ER-bound ATG9A (Fig. 7e), and greatly improved the overt phenotype of the animals (Fig. 7f–i; Supplementary Movie 1). This was also accompanied by a significant increase in both average and maximum lifespan of the mice (Fig. 7i). As expected,

C9 did not rescue the hyperacetylation in the nucleus thus confirming its specificity toward the ER-based ATases (Supplementary Fig. 8a, b).

In conclusion, the above results indicate that restoring the proteostatic functions of the ER by inhibiting ATase1 and ATase2 downstream of AT-1 ameliorates the progeria-like phenotype of SLC13A5 sTg mice. It is also important to stress that for our

**Fig. 4 SLC25A1 sTg and SLC13A5 sTg mice show widespread changes in stoichiometry of lysine acetylation. a** Venn diagram showing overlap between SLC13A5 sTg and SLC25A1 sTg mice across statistically significant acetylation sites (upper panel) and acetylated proteins (lower panel). **b** Histogram showing the distribution of all stoichiometry of acetylation changes from WT in SLC13A5 sTg and SLC25A1 sTg mice. Data are represented as box and whisker plots: box represents 25th to 75th inner quartile range with middle line denoting the median and the inner square denoting the mean; whiskers represent the interquartile distance with a coefficient of 1.5. **P < 0.005 via the Kolmogorov-Smirnov test. **c** Volcano plots displaying stoichiometry of acetylation changes from WT in SLC13A5 sTg and SLC25A1 sTg mice. Statistically significant sites are shown in purple for SLC13A5 sTg mice and orange for SLC25A1 sTg mice. $P < 0.05$ via Fisher's method with all other sites in gray ($n = 5$ male mice per group). **d** The fold enrichment of KEGG pathways determined from proteins harboring the acetylation sites that were significantly changed from WT mice. Proteins (total of 193) that are in common between SLC13A5 and SLC25A1 sTg mice are shown. The top 10 categories sorted by enrichment score (>5) are shown with a filtered FDR score of 0.05. **e** The fold enrichment of KEGG pathways determined from proteins harboring the acetylation sites that were significantly changed from WT mice. Proteins (total of 406) that are unique to SLC13A5 sTg mice are shown. The top 10 categories sorted by enrichment score (>5) are shown with a filtered FDR score of 0.05. **f** The fold enrichment of KEGG pathways determined from proteins harboring the acetylation sites that were significantly changed from WT mice. Proteins (total of 398) that are unique to SLC25A1 sTg mice are shown. The top 13 categories sorted by enrichment score (>5) are shown with a filtered FDR score of 0.05. **g** Distribution of proteins harboring the acetylation sites that were significantly changed from WT mice according to their Uniprot subcellular annotation. Proteins are arranged by cellular location. $P < 0.05$ via one way ANOVA. **h** Gene-network plots of proteins (total of 193) harboring the acetylation sites that were significantly changed from WT and shared by SLC13A5 and SLC25A1 sTg mice. Plots constructed via an overrepresentation analysis using the GO cellular component function database. The dot size of each network category is scaled by the number of overlapping proteins within the category. The top 8 categories sorted by enrichment score are shown with a filtered FDR score of 0.05.

postmortem analysis of treated animals (see Fig. 7d–h), we used 9 month-old SLC13A5 animals, which correspond to more than triple the lifespan of untreated SLC13A5 sTg mice; thus indicating that the protective effects of C9 were long lasting. A schematic interpretation of the results is shown in Fig. 7j.

## Discussion
In the cytosol, acetyl-CoA is mainly generated by ACLY, which uses citrate and CoA[3]. Citrate, in turn, is obtained through active transport from the mitochondria by SLC25A1, and from the plasma membrane by SLC13A5 (see Fig. 1a). Cytosolic acetyl-CoA has access to the nucleus freely, and to the ER lumen through active transport[6]. In this study, we report that mice with systemic overexpression of the citrate transporters, SLC25A1 and SLC13A5, develop strikingly different phenotypes, and present different metabolic responses to changes in citrate/acetyl-CoA availability. This divergent response was observed in the face of overall similar levels of related metabolites at steady-state. Indeed, both models had similar levels of citrate and acetyl-CoA within the cytosol.

**Same metabolite, different phenotypes**. In contrast to the rather benign phenotype of the SLC25A1 sTg model, SLC13A5 sTg mice developed a severe phenotype that was tightly associated with the overexpression of the transporter. Overexpression at conception caused perinatal mortality, while post-natal overexpression caused a segmental form of progeria with reduced lifespan, skin lesions, osteoporosis, splenomegaly, lymphadenopathy with systemic inflammation, hunched posture, rectal prolapse, and accumulation of senescent cells. The progeria-like phenotype of SLC13A5 sTg mice is consistent with the lifespan extension caused by reduced Slc13a5 activity in lower organisms[38,39].

The phenotype of SLC13A5 sTg mice was partially rescued by inhibiting the ER-based acetyltransferases, ATase1 and ATase2. This would indicate that changes in ER acetylation are responsible, at least in part, for the phenotype. In support of this conclusion, we observed increased levels of AT-1, the ER membrane acetyl-CoA transporter that ensures acetyl-CoA availability within the ER lumen, increased acetylation of ER cargo proteins, and increased acetylation of ATG9A, an important partner of the ER acetylation machinery (reviewed in[3]). Importantly, systemic overexpression of AT-1 also caused a segmental form of progeria[32]. The ER acetylation machinery, namely AT-1, ATase1, and ATase2, has emerged as a novel

mechanism that regulates protein homeostasis (proteostasis) within the ER and secretory pathway[3]. This is achieved by ensuring (i) selection of correctly folded polypeptides so that they can engage the secretory pathway, and (ii) removal of toxic protein aggregates through reticulophagy[3]. The latter function requires fine regulation of the acetylation status of ATG9A, downstream of the ATases[32,40–42]. Inhibition of ATase1 and ATase2 corrected the hyper-acetylation of ATG9 and the proteostatic defects observed in AT-1 sTg mice, and rescued their progeria-like phenotype[32,35]. A similar rescuing effect was elicited in SLC13A5 sTg mice (present study). Therefore, defective proteostasis within the ER and secretory pathway is likely to play an important mechanistic role in the SLC13A5 sTg progeria phenotype.

When comparing the SLC13A5 and SLC25A1 sTg models, the stoichiometry of acetylation data reveled a striking representation of the ribosomal machinery, the ER protein-translational machinery, and the ER quality control machinery within the SLC13A5 sTg model. Of note is also the different acetylation profile of Sec62 and Stt3a that was observed in the SLC13A5 and SLC25A1 sTg models. Sec62 is a component of the Sec61/Sec62/Sec63 translocon complex, which ensures translocation of membrane and secreted proteins across the ER membrane, while Stt3a is the catalytic subunit of the oligosaccharyltransferase (OST) complex, which is responsible for the N-glycosylation of nascent glycoproteins as they emerge from the translocon complex. Both Sec62 and OST are important partners of the ER acetylation machinery. Specifically, Sec62 can engage non-acetylated ATG9A to induce reticulophagy[32,42], while OST can interact with the ATases[43]. Consistent with the above, the acetylation profile of SLC13A5 sTg mice revealed unique changes on ER-based chaperones, such as Calr, Hsp90b1, Hspa8, Hspa5 and Hyou1/Grp170, and ER-based folding enzymes, such as Ppib and Pdia3. These changes are all consistent with the increased acetylation of ER cargo proteins observed in the same animals and the effect of ATase inhibition.

Although they display strong phenotypic similarities, the phenotype of SLC13A5 sTg mice was much more severe than the one observed with AT-1 sTg mice[32,35]. The reason for this remains unknown. It is possible that the effect of AT-1 overexpression in AT-1 sTg mice is limited by the availability of cytosolic acetyl-CoA. Indeed, AT-1 sTg mice displayed reduced levels of acetyl-CoA within the cytosol caused by the increased cytosol-to-ER transport[32]. This limitation is not present in mice with overexpression of SLC13A5, where the combination of

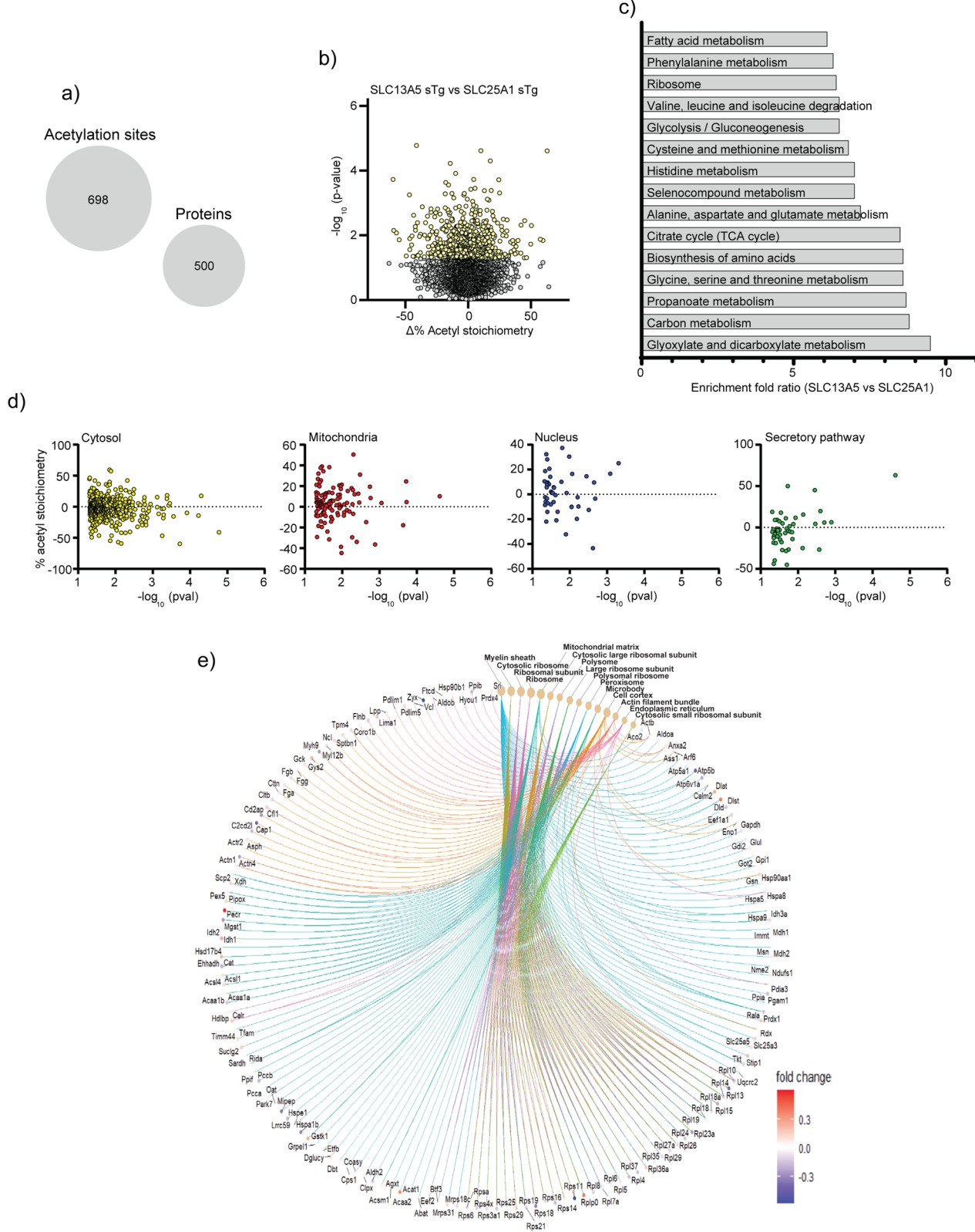

increased endogenous AT-1 and increased availability of acetyl-CoA in the cytosol might be driving ER acetylation beyond the levels observed in AT-1 sTg mice. Unfortunately, this cannot be fully captured at steady-state since acetylated proteins leave the ER with higher efficiency and are deacetylated once they reach the Golgi apparatus[6]. Obviously, it is also possible that increased ER acetylation is only partially responsible for the phenotype of

SLC13A5 sTg mice, and that non-ER events are equally important. Interestingly, CamK2-driven neuron-specific (nTg) overexpression of AT-1, SLC25A1, and SLC13A5 caused an ASD-like phenotype that was overall similar across the three models and remarkably similar in the SLC25A1 nTg and SLC13A5 nTg models[29,30]. In essence, the systemic effects of these three membrane transporters differ from those elicited in the brain.

**Fig. 5 Comparison of SLC13A5 sTg mice to SLC25A1 sTg mice shows unique changes in stoichiometry of lysine acetylation. a** Statistically significant acetylation sites (698) and acetylated proteins (500) unique to SLC13A5 sTg. **b** Volcano plot displaying stoichiometry of acetylation changes that are unique to SLC13A5 sTg mice. Statistically significant sites are shown in yellow. $P < 0.05$ via Fisher's method with all other sites in gray ($n = 5$ male mice per group). **c** The fold enrichment of KEGG pathways determined from proteins harboring the acetylation sites that were significantly changed from SLC25A1 sTg mice. The top 15 categories sorted by enrichment score (>5) are shown with a filtered FDR score of 0.05. **d** Distribution of proteins (total of 500) harboring the acetylation sites that were significantly changed from SLC25A1 sTg mice according to their Uniprot subcellular annotation. Proteins are arranged by cellular location. $P < 0.05$, one way ANOVA. **e** Gene-network plots of proteins (total of 500) harboring the acetylation sites that were significantly changed from SLC25A1 sTg mice. Plots constructed via an overrepresentation analysis using the GO cellular component function database. The dot size of each network category is scaled by the number of overlapping proteins within the category. The top 15 categories sorted by enrichment score are shown with a filtered FDR score of 0.05.

This cell- or tissue-specific divergence might reflect different adaptive responses to the same metabolic signal.

**Same metabolite, different metabolic responses**. The citrate/acetyl-CoA pathway is a recognized key node in metabolism and a powerful regulator of cell homeostasis[2,4,5]. The fact that the increased import of citrate into the cytosol can trigger different adaptive responses depending on where the citrate is coming from, i.e., the mitochondria or the extracellular *milieu*, adds another layer of complexity to our knowledge on how cells respond to fluctuations in nutrient/metabolite availability. Importantly, this divergent response was observed at the level of protein acetylation as well as lipid metabolism.

It is now evident that many proteins and biochemical pathways are directly modulated by Nε-lysine acetylation through the availability and intracellular compartmentalization of acetyl-CoA, donor of the acetyl group for Nε-lysine acetylation[1,5]. This is likely achieved through fluctuations in acetyl-CoA availability occurring within the range of affinity constants of different Nε-lysine acetyltransferases, which enable substrate-mediated regulation of protein acetylation. When analyzed at the stoichiometry level, we found significant changes in Nε-lysine acetylation both at the lysine site (the specific lysine residue) and protein level. These changes were documented in both the SLC25A1 and SLC13A5 sTg models; however, we also observed significant divergence among mouse models.

The measured $K_m$ for citrate transport is in the 650–1400 µM range for human SLC13A5 and 500–750 µM range for human SLC25A1[44–48]. However, citrate export from the mitochondria by SLC25A1 is linked to the import of malate from the cytosol with a measured $K_m$ of about 100–130 µM[44–48]. As a result, the antiporter mechanism is limited by the citrate/malate ratio in both compartments. This limitation is not present with SLC13A5, which is a citrate/$Na^+$ symporter. In light of its transport properties and the concentration of citrate to which it is exposed, SLC13A5 is viewed as a high-capacity transporter[3]. Therefore, different levels of citrate import, and consequent acetyl-CoA availability, may skew the activity of different acetyltransferases, changing the Nε-lysine acetylation profile within different cellular compartments. Interestingly, SLC13A5 sTg mice displayed changes in the acetylation profile of Coasy, a bifunctional protein that carries out the last two biochemical steps in the biosynthesis of CoA. This finding could be consistent with the argument of increased need for CoA. However, the different adaptive response observed in the two sTg models cannot solely be explained by the levels of citrate import (see also Fig. 7j).

About 70% of the sites of acetylation and of the acetylated proteins differed between the two models, highlighting a very different adaptive response imparted upon by the SLC13A5 and the SLC25A1 overexpression (Figs. 4 and 5). This different adaptive response was also evident within the 193 shared proteins where different lysine residues were affected (Supplementary Fig. 2). In essence, the data suggest that the cell is able to

recognize whether the acetyl-CoA, common donor of the acetyl group for acetylation, originates from citrate imported from the mitochondria through SLC25A1, or the extracellular environment through SLC13A5. When analyzed at the pathway level, specific differences also emerged. In general, pathways that feed into the TCA cycle system were uniquely represented in the SLC25A1 sTg model. Highly represented in the SLC25A1 sTg model were also pathways related to fatty acid metabolism and fatty acid oxidation, which might reflect the increased levels of β-hydroxybutyrate and the identification of the peroxisome proliferator-activated receptor pathway among the annotated categories. In contrast, pathways related to the metabolism and biosynthesis of amino acids as well as translation-dependent events, which probably reflect general changes in protein biosynthesis, were abundantly represented in the SLC13A5 sTg model. These include the general ribosomal machinery, the protein translational machinery, as well as the quality control machinery (already discussed above). Also uniquely represented in the SLC13A5 sTg model were pathways related to the elimination of xenobiotics. Importantly, proteins involved with the metabolism of acetyl-CoA/acyl-CoA intermediates within the peroxisomes, as well as proteins involved with oxidative phosphorylation, phosphate and ATP/ADP transfer, and the general proton pump machinery within the mitochondria, were abundantly represented in the acetylome of SLC13A5 sTg mice. Together, these findings strengthen the argument for a unique metabolic response to the overexpression of SLC13A5 and SLC25A1. Unfortunately, large-scale pathway analysis does not inform on the directionality of specific biochemical events; therefore, the biological significance or the specific mechanism(s) behind the divergent response remains to be fully dissected.

As mentioned above, fluctuations in acetyl-CoA availability occurring within the range of affinity constants of different Nε-lysine acetyltransferases enable substrate-mediated regulation of protein acetylation[1,3,5]. As a result, changes in acetyl-CoA levels are not immediately mirrored by similar changes in lysine acetylation of every protein and site. In other words, under conditions where acetylation is dramatically increased on certain proteins and sites, there are other proteins and sites that show a simultaneous decrease[29,30,34,49,50]. This behavior is clearly manifested at the level of the acetyl-proteome where, within the same mouse model and tissue, we observed both increased and decreased stoichiometry of acetylation (see Figs. 4, 5, Supplementary Figs. 2 and 4). This differential response occurred in the face of a general trend toward increased acetylation (see Fig. 4, Supplementary Figs. 2 and 4). In other words, the cell can direct the adaptive response to the increased availability of citrate and acetyl-CoA towards certain biological and biochemical pathways.

The lipidomic analysis also revealed significant divergence between SLC13A5 and SLC25A1 sTg mice. In general, lipid species that are integral components of biological membranes such as phosphatidylethanolamine or phosphatidylcholine were affected by the SLC13A5 manipulation, while lipids that are more

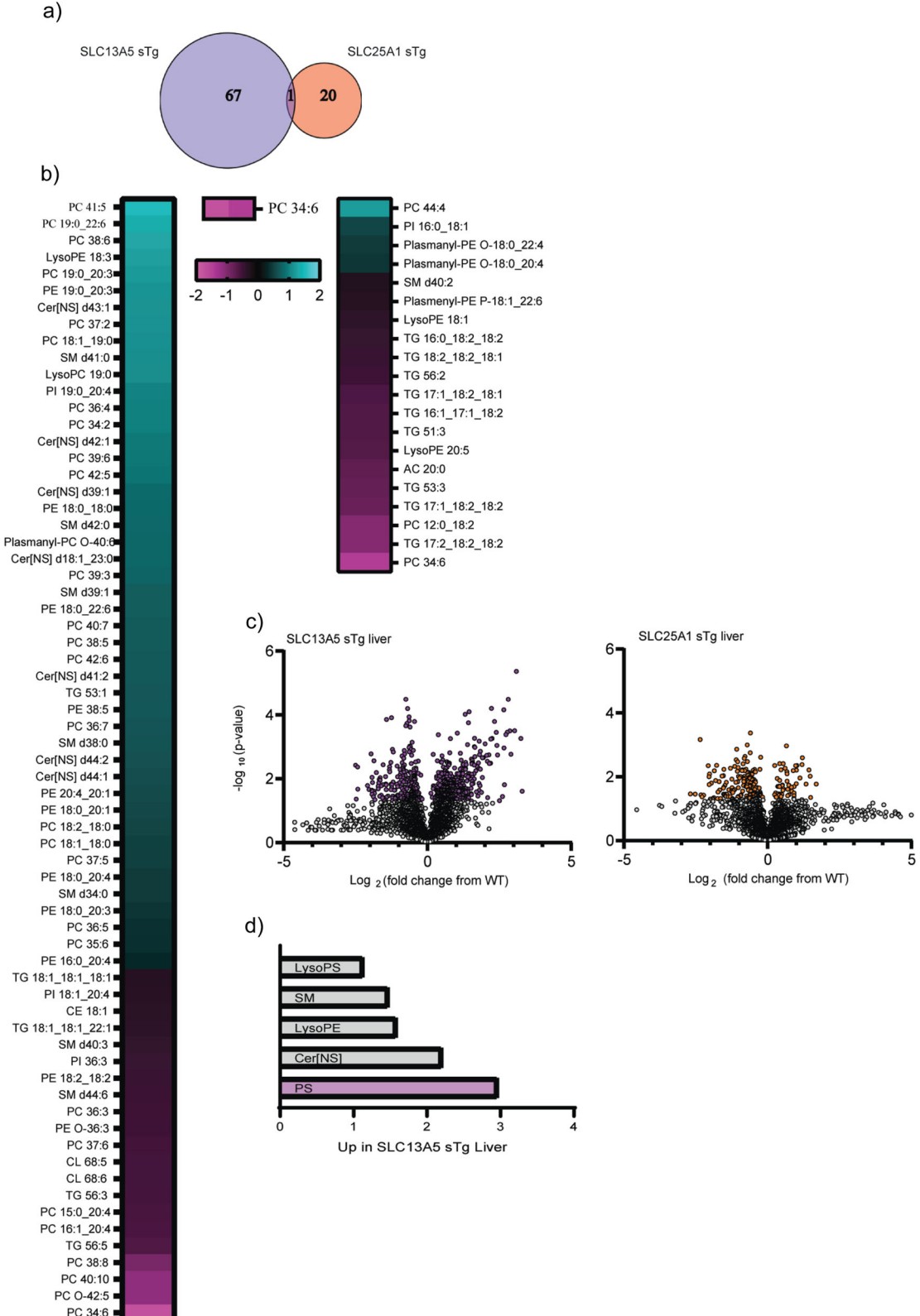

**Fig. 6 SLC13A5 sTg and SLC25A1 sTg mice display different lipid adaptation in the liver. a** Venn diagram showing overlap between SLC13A5 sTg and SLC25A1 sTg mice across statistically significant lipids (*n* = 5 male mice per group). **b** Heat-map showing the expression profile of dysregulated lipids found in SLC13A5 sTg and SLC25A1 sTg mice as compared to WT littermates. **c** Volcano plot displaying all quantified lipids in SLC13A5 sTg and SLC25A1 sTg livers, as compared to age-matched WT littermates. Statistically significant species are highlighted in purple (for SLC13A5 sTg) and orange (for SLC25A1 sTg). *P* < 0.05 via Fisher's method with all other species in gray. **d** Bar graph representing up-regulated groups of lipids in SLC13A5 sTg liver samples. Statistically significant groups represented in purple.

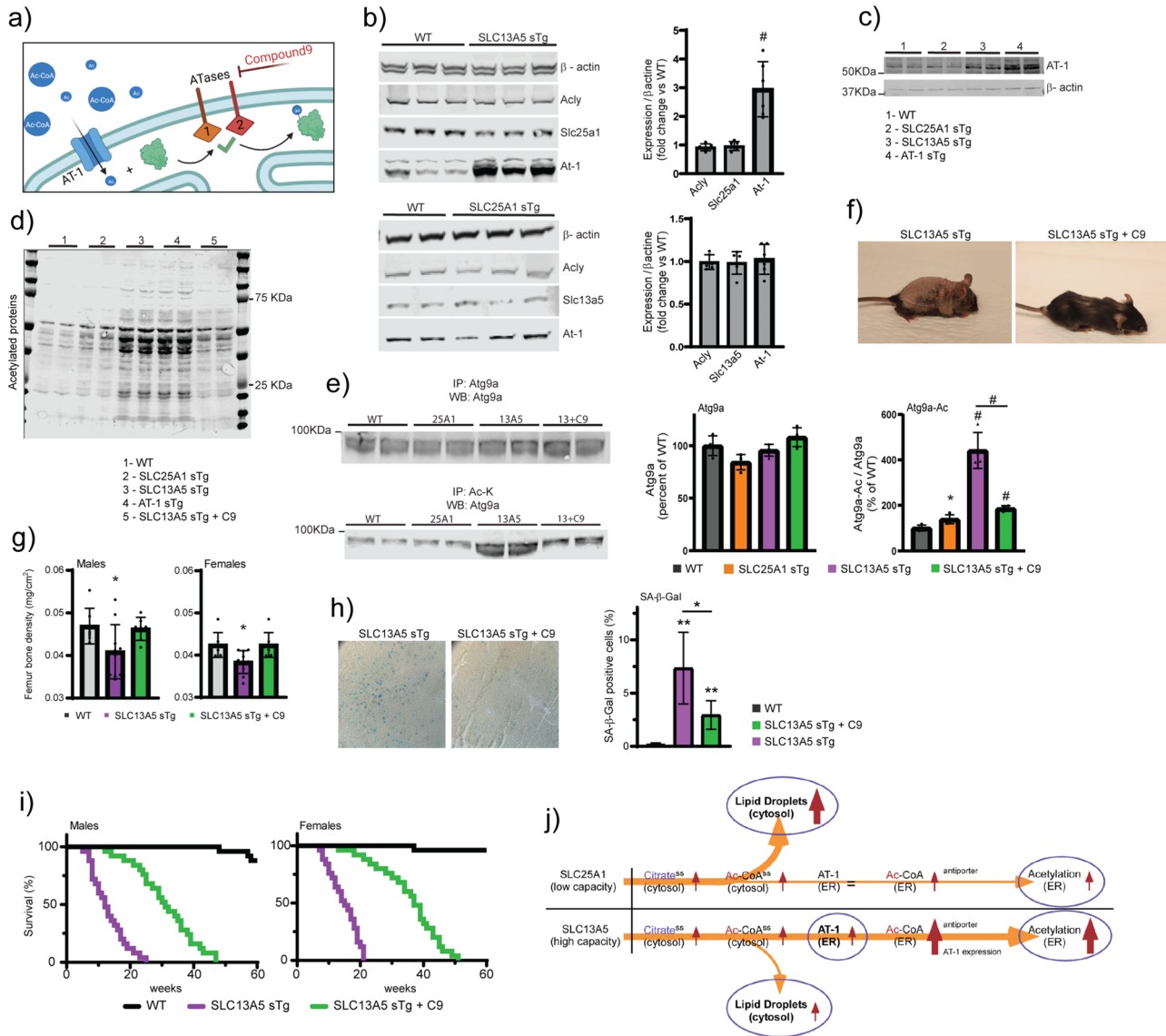

**Fig. 7 ATase1/ATase2 inhibition rescues the progeria-like phenotype and lifespan of SLC13A5 sTg mice. a** Schematic view of the ER acetylation machinery with compound 9 acting on the ATases downstream of AT-1. **b** Western blot showing expression of different target proteins in SLC13A5 sTg and SLC25A1 sTg mice. Representative Western blot (left) and quantification (right) are shown. #$P < 0.0005$ via mean comparison using Student's $t$ test ($n = 6$ male mice per group). **c** Western blot showing expression of AT-1 and β-actin proteins. **d** Western blot showing the profile of Nε-lysine acetylation in the ER isolated from liver. **e** Western blot showing the acetylation profile of ER-bound Atg9a. Representative Western blot (left) and quantification (right) are shown (Atg9a, total Atg9a; Atg9a-Ac, acetylated Atg9a). **f** Representative images of SLC13A5 sTg with and without C9 treatment. Mice were 120 days old. **g** Femur bone density from SLC13A5 sTg animals with and without C9 treatment. *$P < 0.05$ via mean comparison using Student's $t$ test (WT, $n = 7$; SLC13A5 sTg, $n = 10$; SLC13A5 sTg + C9, $n = 7$). **h** SA-β-Gal staining of liver slides from SLC13A5 sTg animals with and without C9 treatment. Representative images and quantification of results are shown. *$P < 0.05$ via mean comparison using Student's $t$ test ($n = 4$ mice/group; 150 cells/animal). **i** Lifespan of SLC13A5 sTg mice with and without C9 treatment ($n = 25$/group). $P < 0.0005$ via Kaplan–Meir lifespan test. **j** Schematic summary of the different adaptive response elicited by overexpression of SLC25A1 and SLC13A5 (ss, steady state).

typically stored in the cytosol such as triglycerides were affected by the SLC25A1 manipulation. As with the stoichiometry of acetylation, we did not observe a generalized increase in lipid levels but rather a divergent response at the class, sub-class, and specie level. This divergent response was also very different across the tissues examined (see Fig. 6 and Supplementary Fig. 5–7). Interestingly, the lipidomic analysis revealed fewer changes in the brain as compared to the other tissues, perhaps suggesting that lipogenic substrates are already optimized and less responsive to substrate fluctuations in the brain. Indeed, the most salient finding in the brain was the marked reduction in glyco-sphingolipids and sulfatides within the SLC25A1 sTg model,

perhaps indicating alterations in myelin formation/composition. Although the biochemical and biological significance of the very different lipid profiles observed in the models remain unknown, it is clear that SLC13A5 and SLC25A1 elicit very different metabolic responses, which appear to mirror the differences observed within the stoichiometry of acetylation data.

**Conclusions**

Overall, our results support the conclusion that the intracellular flux of citrate and acetyl-CoA, as regulated by the interplay of different membrane transporters, might influence unique

organelle-specific adaptive responses. This intracellular crosstalk does not simply depend on the levels of the individual metabolites but also on the source of the metabolic intermediate, which would allow the cell to tune the response to the specific metabolic signal/event. Finally, our results indicate that SLC13A5-derived citrate is tightly linked to ER proteostasis and associated progeria, while SLC25A1-derived citrate is not.

## Methods

**Transgenic mouse generation**. Rosa26:tTA;pTRE-SLC13A5 (referred to as SLC13A5 sTg) and Rosa26:tTA;pTRE-SLC25A1 (referred to as SLC25A1 sTg) mice were generated by crossing Rosa26:tTA mice with pTRE-SLC13A5 and pTRE-SLC25A1 mice, respectively. The generation of Rosa26:tTA mice is described in[32], while the generation of pTRE-SLC13A5 and pTRE-SLC25A1 mice is described in[29,30]. Genotyping from tail DNA was performed at weaning by TransnetYX using real time PCR[29,30,32].

**Animals**. All the animals used in this study are *Mus musculus* strain C57BL/J6. Mice were housed in standard cages provided by the University Laboratory Animal Resources and grouped with littermates, 1–5 per cage; animals were supplied standard chow and water *ad libitum*. The diet with Compound 9 was manufactured by Bio-Serv[32,35]. The diet with doxycycline (200 mg/kg) was purchased from Bio-Serv[51]. The same diet without Compound 9 or doxycycline served as the control diet. All animal experiments were performed in accordance with the National Institute of Health Guide for the Care and Use of Laboratory Animals and were approved by the Institutional Animal Care and Use Committee of the University of Wisconsin–Madison (protocol #M005120). Non-transgenic littermates were used as controls throughout the study. All studies were conducted with 2–4 month-old animals, which corresponded to the humane-end point for SLC13A5 sTg mice. Only exception were C9-treated SLC13A5 sTg mice, which were used at the age of 9 months to determine long-lasting effects of the therapeutic rescue.

**Western blotting**. Western blotting was conducted as previously described[33,37,41,52,53]. The following primary antibodies were used in this study: α-actin (Cell signaling #4976L), SLC25A1 (Proteintech #15235-1-AP), SLC13A5 (Santa Cruz #sc293277), AT-1 (Aviva System Biology #ARP43888_P050), Acetylated Lysine (Cell Signaling #cs9441L), ACLY (Abcam #ab40793), Calnexin (Novus #NB100-1974). Donkey anti-rabbit or goat anti-mouse IRDye 800CW and 680RD-conjugated secondary antibodies (LI-COR Biosciences, #926-32213, #926-32210, #926-68073, #926-68070) were used for infrared imaging (LI-COR Odyssey Infrared Imaging System; LI-COR Biosciences). For enriched liver ER and nuclear Western blotting, target proteins were normalized to total protein staining with the Revert Total Protein Stain (LI-COR Biosciences, #926-11021) performed before immunodetection. Original uncropped Western blot images included in the manuscript can be found in Supplementary Fig. 9.

**Organelle enrichment and immunoprecipitation**. Total ER isolation was prepared using a commercially available ER Enrichment kit (Novus Biologicals, #NBP2-29482), according to the manufacturer protocol (see also[32]). Briefly, 0.5 g of liver tissue was homogenized in isosmotic homogenization buffer using a Dounce Teflon homogenizer for 20–30 strokes. The homogenized tissue was centrifuged at $1000 \times g$ for 10 min at 4 °C; the supernatant was centrifuged again at $12,000 \times g$ for 15 min at 4 °C, discarding the pellet afterwards. Finally, the supernatant was

centrifuged at $90,000 \times g$ for 60 min at 4 °C to obtain total ER fraction as a pellet. Nuclear protein extracts were prepared using the NE-PER kit (ThermoFisher Scientific, #78833) from fresh liver samples according to protocol instructions. The mitochondria fraction was prepared as previously described[54].

For immunoprecipitation, the total ER pellet was resuspended in kit-provided 1X suspension buffer supplemented with protease inhibitor cocktail. Immunoprecipitation was performed with magnetic beads (Bio-Rad; #161-4013) and 500–1000 μg enriched liver ER using anti-acetylated lysine (Cell Signal Technologies; #9441; 1:100) or anti-ATG9A (Abcam; #ab108338; 1:100) primary antibodies.

**Acetyl-CoA, citrate and malate assays**. Mice were fasted by removing diet at the beginning of the light cycle (06:00), and were $CO_2$-euthanized after 6 h (12:00). Liver was rapidly removed and diced into small pieces, rinsed in PBS, and flash frozen. Enriched liver subcellular fractions were deproteinized (BioVision; #K808) according to kit instructions. Specific assays were used to determine the concentrations of acetyl-CoA (Abcam Acetyl-CoA Assay Kit #ab87546), citrate (Abcam Citrate Assay Kit #ab83396) and malate (Abcam Malate Assay Kit #ab8391). Assays were run according to kit instructions.

**Histology and immunostaining**. For H&E staining, the individual organs were collected and immediately placed in 10% neutral buffered formalin overnight, embedded in paraffin using standard techniques, and then sliced on a microtome. Sections underwent deparaffinization and rehydration, and were processed for H&E staining. Processed slides were imaged on a Zeiss Axiovert 200 inverted fluorescent microscope.

The Liver LipidTox staining was performed as described[49]. Specifically, liver sections were collected, immediately placed in the Optimal Cutting Temperature medium (ThermoFisher Scientific, San Jose, CA, USA), and stored at −80 °C until processed. Sections were sliced on a microtome cryostat (Microm HM 505N), and processed using the HCS LipidTOX™ Neutral Lipid Stains protocol (ThermoFisher). Processed slides were then imaged on a Zeiss Axiovert 200 inverted fluorescent microscope.

For immunofluorescence analysis, tissues were deparaffinized, re-hydrated, and processed for heat-induced epitope retrieval. Epitopes were blocked in 10% goat serum for 3 h at room temperature. Samples were incubated overnight with the primary antibody, then washed and incubated for 1 h with secondary antibodies. Samples were visualized on a confocal microscope (Nikon A1RS Confocal Microscope). For quantification of mitotic index and measurement of skin basal layer, six independent fields were obtained and quantified using open-source Fiji image processing package (https://loci.wisc.edu/software/fiji). Antibodies and immunohistochemical reagents were as follows: anti-Ki67 (BD Biosciences, #550609), anti-Keratin 5 (Bio-Legend, #PRB-160P), anti-Keratin 10 (Bio-Legend, #PRB-159P); secondary reagents were AlexaFluor546 goat anti-rabbit (#A-11035) and AlexaFluor700 goat anti-mouse (#A-21036) from Thermo Fisher Scientific. Nuclei were stained by mounting in ProLong Gold Antifade Reagent with DAPI (Life Technologies, #P36935).

**Reverse transcription quantitative PCR (RT-qPCR)**. RNA extraction, cDNA synthesis, and reverse transcription (RT-qPCR) were performed as previously described[32,33,49]. Real-time PCR was performed using the Roche 480 lightcycler and SYBR Green Real Time PCR Master Mix (Roche, #04707416001). Expression levels were normalized against GAPDH levels, and expressed as fold change vs control. The following primers were used: p21 forward (5′-GTG ATT GCG ATG CGC TCA TG-3′), p21 reverse

(5′-TCT CTT GCA GAA GAC CAA TC-3′), p16 forward (5′-CCC AAC GCC CCG AAC T- 3′), p16 reverse (5′-GCA GAA GAG CTG CTA CGT GAA-3′), GAPDH forward (5′-AGG TCG GTG TGA ACG GAT TTG- 3′), and GAPDH reverse (5′-TGT AGA CCA TGT AGT TGA GGT CA-3′).

**Stoichiometry of acetylation**. Stoichiometry of acetylation was analyzed as before[29,30,34,49]. Briefly, 50 μg of protein from the liver mitochondrial/cytosolic, nuclear pool, and chromatin-associated subcellular fractions were denatured in urea, alkylated with iodoacetamide, chemically acetylated using isotopic D6-acetic anhydride. The peptides were treated with two rounds of labeling with D6-acetic anhydride to achieve >99% labeling of lysine residues. Then the peptides were digested with a combination of Trypsin and GluC. The samples were analyzed using data-independent acquisition (DIA) analysis by a Thermo Q-Exactive Orbitrap coupled to a Dionex Ultimate 3000 RSLC nano UPLC with a Waters Atlantic reverse phase column (100 μm × 150 mm). To deconvolute and analyze the DIA spectra, a spectral library was generated. Spectral library samples were processed identically to the experimental samples, except they were treated with non-isotopic acetic anhydride. Peptides were pre-fractionated into six fractions using offline, high pH reverse phase chromatography prior to cleanup and desalting for mass spectrometry analysis. They were then analyzed using data dependent acquisition (DDA) mass spectrometry analysis.

Using the openly available MaxQuant (v1.6.1) software package, a database search was performed to identify peptides present in the DDA samples analyzed. Carbamidomethylation (C) was set as a fixed modification, and Oxidation (M) and Acetyl (K) were set as variable modifications. Trypsin and GluC were set as the digestion enzymes, with the maximum number of missed cleavages set to five. DDA runs from all subcellular fractions were combined to generate one combined library. Heavy acetyl fragment ion pairs were generated in silico, such that the spectral library would contain both the light (endogenous) acetylation peaks and the heavy (chemical) acetylation peaks.

The experimental samples were processed with Spectronaut Pulsar. The data were processed using an in-house R script[49], such that stoichiometry was calculated from the ratio of endogenous (light) fragment ion peak area over the total (endogenous and chemical) fragment ion peak area. Isotopic envelope correction of the heavy labeled peak was performed to remove any contribution from naturally occurring isotopes from the light labeled peak. To identify acetylation sites that were significantly different between the overexpression models and WT, we calculated p-values using linear modeling and one-way analysis of variance (ANOVA).

**Lipidomic analysis**. Individual samples were pulverized by mortar and pestle under liquid nitrogen. Approximately 15 mg (frozen wet weight) material was transferred into microcentrifuge tubes; wet weight was recorded and used for normalization. Tissue samples were maintained frozen until the time of extraction. To each 15 mg sample, 500 μL of ice cold extraction solvent was added; extraction solvent consisted of 6:2:2 (v/v/v) n-Butanol:Acetonitrile:Water[55]. A 5 mm steel bead was added to each sample, and the sample was subjected to 5 min of bead beating on a Retsch MM400 at a frequency of 25 hz at 4 °C. Samples were then centrifuged at 14,000 × g for 2 min at 4 °C. Next, 100 μL of extract was transferred to an amber glass autosampler vial with fused glass insert. This extract was dried by vacuum centrifugation for approximately 1 hr. Samples were resuspended in 50 μL of 9:1 methanol:toluene for analysis by LC-MS.

For LC-MS lipidomics analysis, 10 μL of extract was injected by a Vanquish Split Sampler HT autosampler (Thermo Scientific) onto an Acquity CSH C18 column held at 50 °C (100 mm × 2.1 mm × 1.7 μm particle size; Waters) using a Vanquish Binary Pump (200 μL/min flow rate; Thermo Scientific). Mobile phase A was 0.2% formic acid in water, and mobile phase B (MPB) was 0.2% formic acid in IPA:ACN (90:10, v/v) with 10 mM ammonium formate. The gradient was as follows: 5% MPB for 1 min and then over 3 min increase to 40% MPB. Over the next 11 min increase to 70% MPB, then over 4 min increase to 80.5% Mobile Phase B, next 2 min increase to 85% Mobile Phase B, next 5 min increase to 89.5% MPB, next 3 min increase to 91% MPB, next 5 min increase to 100% MBP, then hold at 100% MPB for 5 min. The column was re-equilibrated with to 5% MBP for 9 min[55].

The LC system was coupled to a Q Exactive HF Orbitrap mass spectrometer. The column eluate was ionized using a heated electrospray ionization (HESI II) source (Thermo Scientific), with 275 °C vaporization and capillary temperature, 30 unit sheath gas, 6 unit aux gas, 0 unit sweep gas, |4.5 kV| spray voltage for both positive and negative modes, and 60 unit S-lens RF settings. The MS was operated with both positive and negative full MS and MS2 spectra (Top2) within the same injection. Full MS scans were acquired with 30,000 resolution, $1 \times 10^6$ automatic gain control (AGC) target, 100 ms ion accumulation time (max IT), and 200 to 1600 m/z scan range. MS2 scans were acquired at 30,000 resolution, $1 \times 10^5$ AGC target, 50 ms max IT, 1.0 m/z isolation window, stepped normalized collision energy (NCE) at 20, 30, 40, and a 30.0 s dynamic exclusion.

The resulting raw data were processed with Thermo Scientific software Compound Discoverer 3.1 and LipiDex[31]. Peak detection was set between 1 and 41 min retention time. Peak alignment maximum shift was set to be 1.2 min. Features were aggregated into compound groups if they exceeded $1 \times 10^5$ minimum peak intensity, were less than 0.75 maximum peak width, had minimum 3 signal-to-nosie ratio, and were 3-fold over blanks. MS/MS spectra were searched using "LipiDex_HCD_Formic" library with C2:C12 acyl chains checked. MS/MS with spectral matches having dot product score >500 and reverse dot product scores >700 were retained. Minimum spectral purity was set to 75% for designating fatty acid composition. Features were required to be found in >2 files. Tissue samples were normalized to the wet weight extracted.

**Senescence β-galactosidase assay**. Cryosections of mouse liver were stained with Senescence β Galactosidase Staining kit (Cell Signaling Technology, #9860S) according to the manufacturer's protocol. Briefly, mouse liver cryosections (10 μm) were fixed with fixation solution for 15 min. Fixed sections were then stained with β-galactosidase at 37 °C overnight in a dry incubator without carbon dioxide. The percentage of senescent cells was expressed as the total number of stained senescent cells divided by the total number of cells.

**Bone mineral density**. Bones were fixed in 70% ethanol and soft tissues were removed from the bone once fixed. Bone mineral density (BMD) was determined using the UltraFocus DXA system (Faxitron) following standard manufacturer protocols[35].

**Transmission electron microscopy**. Following $CO_2$ euthanasia, liver was extracted and fixed in 2.5% glutaraldehyde in 0.1 M phosphate buffer (PB) overnight at 4 °C. Fixed samples were rinsed 5 × 5 min in PB and post-fixed in 1% osmium tetroxide, 1% potassium ferrocyanide in 0.1 M PB for 1 h at room temperature, then rinsed in PB. Dehydration was performed in

ethanol series (35%, 50%, 70%, 80%, 90% for 10 min each step, 95% for 20 min, 100% for 2 × 10 min) at room temperature and 100% ethanol at 4 °C overnight, then transitioned in propylene oxide (PO) 2 × 7 min at room temperature. Fully dehydrated samples were infiltrated in increasing concentrations of PolyBed 812 (Polysciences Inc.) and PO mixtures. Embedding and polymerization occurred in fresh PolyBed 812 for 24 h at 60 °C. The samples were sectioned on a Leica EM UC6 ultra-microtome at 100 nm, collected on formvar-coated 2 × 1 mm slot Cu grids (EMS Hatfield, PA) and post-stained with uranyl acetate and lead citrate. The sectioned samples were viewed at 80 kV on a Philips CM120 transmission electron microscope equipped with AMT BioSprint12 digital camera (AMT Imaging Systems).

**Dot blots**. Fractions enriched in extracellular (EC), intracellular (IC), or membrane-associated (MB) proteins were prepared as previously described[32]. Briefly, liver tissue was homogenized in EC buffer (50 mM Tris-HCl, pH 7.6; 150 mM NaCl; 2 mM EDTA; 0.01% (w/v) SDS; and 0.01% (v/v) NP-40) with inhibitors. After centrifugation at 13,200 rpm for 90 min at 4 °C, supernatants containing extracellular proteins were collected. Pellets were then homogenized in IC buffer (50 mM Tris-HCl, pH 7.6; 150 mM NaCl; and 0.1% (v/v) TritonTM X-100; with inhibitors) and re-centrifuged. Resulting IC supernatants were collected, and the remaining pellets were nutated (15 min, 4 °C) in MB buffer (50 mM Tris-HCl, pH 7.4; 150 mM NaCl; 1 mM EGTA; 3% (w/v) SDS; 1% (w/v) deoxycholate; and 0.5% (v/v) TritonTM X-100; with inhibitors). Samples were then centrifuged at 13,200 rpm for 90 min at 4 °C and MB supernatants were collected.

One microgram of liver extract enriched in either extracellular, intracellular, or membrane-associated proteins was diluted in PBS to a total volume of 3 μL, then dotted onto a nitrocellulose membrane (Bio-Rad), and allowed to dry. Once dry, membranes were rinsed twice in TBS and incubated for one hour at room temperature in 5% BSA (Sigma-Aldrich) in TBST. Membranes were then incubated overnight at 4 °C in 5% BSA in TBST, washed four times in TBST, and incubated for one hour at room temperature in TBST with IRDye 800CW conjugated goat anti-mouse secondary antibody (LICOR Biosciences, #926-32210). After secondary incubation, membranes were washed four additional times in TBST, rinsed once in TBS, and then imaged using the LI-COR Odyssey Infrared Imaging System.

**Blood ketone and glucose determinations**. Mice were fasted by removing the diet at the beginning of the light cycle (06:00), and were CO2-euthanized after 6 h (12:00). Blood was rapidly collected using intracardiac injection. Glucose and β-hydroxybutyrate were measured using fresh blood with a nova-Vet Blood Ketone and Glucose Monitoring System (Nova Biomedical, #53574; glucose test strips, #53069; ketone test strips, #52587).

**Statistics and reproducibility**. No statistical method was used to determine the necessary sample size for each experiment. The number of experimental replicates, representing the number of mice per genotype, is indicated in the respective legends. A minimum of $n = 3$ animals and a maximum of $n = 25$ animals were used throughout our study as required by the experimental design. Data analysis was performed using GraphPad Prism version 9.0.2. Data are expressed as mean ± standard deviation (SD) unless otherwise specified. Comparison of the means was performed using Student's t-test for two groups. Statistical test details are described in the figure legends. Differences were declared statistically significant if $P < 0.05$ and the following statistical significance indicators are used: *$P < 0.05$; **$P < 0.005$; #$P < 0.0005$.

**Reporting summary**. Further information on research design is available in the Nature Portfolio Reporting Summary linked to this article.

## Data availability

The acetyl-proteomics data that support the findings of this study have been deposited to the ProteomeXchange Consortium (ID: PXD040008) and the MassIVE partner repository (ID: MSV000091235). The lipidomic data that support the findings of this study have been deposited to the MassIVE repository (ID number MSV000090972). Source data for the graphs and charts are available as Supplementary Data 1 and any remaining information can be obtained from the corresponding author upon reasonable request.

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

## Acknowledgements

This research was supported by National Institutes of Health (R01NS094154, R01GM148487, R01AG078794, R01AG057408, R01GM065386, P30DK020579, and P41GM108538), and a core grant to the Waisman Center from NICHD-U54 HD090256. We would like to thank Kathy Krentz at the Genome Editing & Animal Models Core of UW-Madison for the generation of the SLC25A1 and SLC13A5 transgenic mice, and Ruth Sullivan for the pathological analysis of embryo tissues. We are also grateful to John Svaren and Alan Attie for their evaluation of an early version of this manuscript.

## Author contributions

Conceptualization, G.F.-F. and L.P.; methodology, investigation, data curation, formal analysis and visualization, G.F.-F, K.A.O, A.J.L., I.K. and S.L.S; formal analysis, P.G.-M., J.J.C, J.M.D., C.M.A. and L.P.; supervision, L.P.; project administration, L.P.; writing of original draft, G.F.-F., K.A.O., A.J.L., I.K. and L.P.; review and editing, all authors.

## Competing interests

The authors declare the following competing interests: J.M.D. is a co-founder of Galilei BioScience Inc. and a consultant for Evrys Bio. J.J.C. is a consultant for Thermo Fisher Scientific, 908 Devices, and Seer. The remaining authors have no competing interests to disclose.
