## [Peer Review File · Communications Biology]

Reviewers' comments:

Reviewer #1 (Remarks to the Author):

In this paper the Authors investigated the role played by the citrate transporters SLC13A5 and SLC25A1 inside the cells.

The topic of the study is very intriguing, because it becomes progressively clearer that citrate transport proteins do not play just the role of transporter of citrate across the lipid bilayer, but they are fully involved in many cellular processes. The understanding of the role played by the citrate transporters SLC13A5 and SLC25A1 in cellular metabolism and physiology can provide useful insight into the molecular basis of the human pathologies in which these carrier proteins are involved.

The article is well written, and the study is well executed.

I suggest only to introduce two aspects in the discussion section:

1) Recent studies suggest that exogenous citrate transported by SLC13A5 appreciably contributes to intermediary metabolism only in oxygen- and glutamine-limited conditions. How this evidence could be linked to the acetylation machinery?

2) The citrate export mediated by SLC25A1 is coupled to the entry of cytosolic malate into mitochondria. In the mitochondrial matrix, malate enters the Krebs cycle generating NADH, which, in turn, donates the reducing equivalents to complex I of the respiratory chain, thereby stimulating OXPHOS. How the increase in the mitochondrial malate levels could be linked to phenotypes associated to SLC25A1 overexpression?

Reviewer #2 (Remarks to the Author):

SLC13A5 and SLC25A1 are two important citrate transporters. Many studies have reported pathological conditions caused by deletion of these two genes, it involves neurological disorders, myasthenia, 2-hydroxyglutaric aciduria and so on. This paper systematically describes the phenotype of mice with systemic overexpression of SLC13A5 and SLC25A1, just as Table 1 is listed. The phenotypes of SLC13A5 and SLC25A1 sTg mice are very strong and significant, there are many phenotypes that deserve further research. It is a novel finding that SLC13A5 overexpression mice have a progeria-like phenotype, although it doesn't explain much about the mechanism, and SLC13A5 may be a drug target for some rare diseases. In this paper, Nε-lysine acetylation profiles and lipid profiles of two types of mice were compared using omics methods, it reveals that different sources of citrate may have different metabolic responses, this also illustrates the complexity of citrate metabolism. Here are some questions for you.

1. Citrate is an inhibitor of 6-phosphofructokinase1, which is the rate-limiting enzyme for glycolysis. Whether SLC25A1 sTg and SLC13A5 sTg mice show inhibition of glycolysis ? Whether it is related to the phenotype of SLC25A1 sTg and SLC13A5 sTg mice ?
2. Different sources of citrate have different phenotypes in mice. What are the possible hypotheses to explain it ?
3. Although SLC13A5 mainly transports citrate, it also transports other organic acids and sodium, these metabolites may develop a progeria-like phenotype of SLC13A5 sTg mice. Can you explain why it's not these metabolites that cause the phenotype ?
4. Many foods contain citrate, whether SLC13A5 sTg mice's phenotype can be rescued or get worse by citrate supplement ?
5. It's better to add the position of marker when western blotting is full film exposure, such as Figure 7D.
6. What will happen when both SLC13A5 and SLC25A1 are overexpressed in mice ?

7. Sometimes β -Actin might have two strips, more often there is only one strip, such as Figure 7B (SLC25A1 sTg) and Figure 7C. Why are there two strips in Figure 7B (SLC13A5 sTg) ? Are there always two strips in this kind of tissue ?
8. Loss of SLC13A5 has been reported to be associated with epilepsy, whether SLC13A5 overexpression can improve related phenotypes or indicators ? Can you evaluate the potential of SLC13A5 as a potential prescription and the potential side effect of premature aging ?
9. In general, increased lipid accumulation in the liver tends to mean heavier liver weight. Can you explain why SLC25A1 sTg mice have more lipid droplets, but less weight of liver ? Could this be caused by a loss of total body weight of SLC25A1 sTg mice ?
10. SLC13A5 sTg mice display a progeria-like phenotype, in this paper you have used some experiments to describe it, for SLC25A1, why did its overexpression reduce the size of mice so much in Figure 1C ? What kind of disease might be involved ?

Reviewer #3 (Remarks to the Author):

Puglielli et al. investigated the distinct roles of SLC25A1 or SLC13A5 in the import of citrate and related acetylation changes using transgenic mice. The experiment was designed and executed with a clear rationale, and the data was presented appropriately. However, the mechanistic insights, revealed by this study, are limited, and the proposed mechanisms are not supported by protein acetylation data. There are some critical questions, particularly regarding the acetylomic data, that need to be addressed before the study can be acceptable for publication in Communications Biology.

1. Why did the authors choose to measure acetylation stoichiometry instead of relative change of acetylation abundance? Acetylation is widely recognized as an extremely low stoichiometric event, as most acetylation occurs at very low stoichiometry (median 0.02%), high stoichiometry acetylation (>1%) only occurs on nuclear proteins involved in gene transcription and on acetyltransferases (PMID: 30837475). The regular measurement error of MS intensity is already much larger than 1%, making the accuracy of stoichiometry measurements, particularly after subcellular fractionation, likely to be very poor. For example, a 1% acetylation stoichiometry can be easily measured and calculated as 2% (or even larger) in MS quantification, which can generate an artificial 2-fold change in stoichiometry calculation. This is supported by the data presented in Figure 4B and 4C, which show patterns for random changes in stoichiometry rather than the expected increase in transgenic mice. In addition to the fold-change, the original stoichiometric values need to be displayed.
2. It is important to evaluate the reaction efficiency of chemical acetylation using acetic anhydride. Specifically, it is important to determine whether all free lysine residues were acetylated by the isotope-labeled reagent in the proteomics data. If not, it is crucial to calculate the labeling efficiency, as incomplete labeling could lead to significant errors in stoichiometry calculations.
3. It is recommended that the authors examine the total acetylation levels of different tissues in both wild-type and transgenic mice using a pan-anti-acetyl-lys antibody. Ideally, the acetylome data based on antibody enrichment are much helpful, and should be considered.
4. The performance of subcellular fractionation should be assessed by examining subcellular-specific protein markers.

POINT-BY-POINT RESPONSE

We wish to thank the Editor and the Reviewers for their positive comments. Changes within the manuscript are highlighted. A comprehensive point-by-point response to the suggestions and questions can be found below.

Reviewer #1

I suggest only to introduce two aspects in the discussion section:

1) Recent studies suggest that exogenous citrate transported by SLC13A5 appreciably contributes to intermediary metabolism only in oxygen- and glutamine-limited conditions. How this evidence could be linked to the acetylation machinery?

Response: The results mentioned by the Reviewer were obtained with immortalized cancer-derived cell lines in culture (*ex vivo*). We do not know whether there is a link between intermediate metabolism under oxygen-/glutamine-limited conditions and ER acetylation *in vivo* (in the whole animal). Although interesting, this is beyond the scope of this manuscript and is more suitable for discussion within a review paper exploring *ex vivo* vs *in vivo* data.

2) The citrate export mediated by SLC25A1 is coupled to the entry of cytosolic malate into mitochondria. In the mitochondrial matrix, malate enters the Krebs cycle generating NADH, which, in turn, donates the reducing equivalents to complex I of the respiratory chain, thereby stimulating OXPHOS. How the increase in the mitochondrial malate levels could be linked to phenotypes associated to SLC25A1 overexpression?

Response: This is an interesting point that we have targeted (and are still trying to target). However, currently, we do not have any data that would support a differential engagement of the TCA cycle and the respiratory chain between SLC25A1 and SLC13A5 sTg mice. This also means that we cannot use this as a mechanistic argument to explain the different phenotypes of our two mouse models. If the Reviewer is interested, we can certainly provide (confidentially) our unpublished data.

Reviewer #2

Here are some questions for you.

1. Citrate is an inhibitor of 6-phosphofruktokinase1, which is the rate-limiting enzyme for glycolysis. Whether SLC25A1 sTg and SLC13A5 sTg mice show inhibition of glycolysis ? Whether it is related to the phenotype of SLC25A1 sTg and SLC13A5 sTg mice ?

Response: We have no evidence that an immediate effect on glycolysis is related to the phenotype of SLC25A1 sTg or SLC13A5 sTg mice.

2. *Different sources of citrate have different phenotypes in mice. What are the possible hypotheses to explain it ?*

Response: The main argument is the differential effect on AT-1 and ER acetylation. SLC13A5 sTg mice displayed (i) marked upregulation of endogenous AT-1 (Fig. 7); (ii) increased acetylation of ER cargo proteins (Fig. 7); (iii) increased acetylation of ATG9A (Fig. 7); (iv) generalized changes (as revealed by unbiased mass spectrometry) of the secretory pathway (Fig. 5). These changes were uniquely observed in SLC13A5 sTg mice. Furthermore, the inhibition of the ATases was able to rescue the hyperacetylation status of the ER and ATG9A, and ameliorate the phenotype (Fig. 7). The fact that overexpression of AT-1 (AT-1 sTg mice; PMCID: PMC6156544 and PMCID: PMC8881600), ATase1 (ATase1 sTg mice; currently unpublished), and ATase2 (ATase2 sTg mice; currently unpublished) also causes similar forms of segmental progeria is an additional strong argument in favor of the mechanistic involvement of ER acetylation in the progeria phenotype of SLC13A5 sTg mice. The mechanistic aspects of the SLC13A5 sTg phenotype are already discussed in the manuscript. The Discussion section also elaborates on the complexity of the SLC13A5 sTg and SLC25A1 sTg phenotypes.

3. *Although SLC13A5 mainly transports citrate, it also transports other organic acids and sodium, these metabolites may develop a progeria-like phenotype of SLC13A5 sTg mice. Can you explain why it's not these metabolites that cause the phenotype ?*

Response: It is true that *in vitro* (reconstituted proteoliposomes), mammalian SLC13A5 can transport other monocarboxylates. However, the relative K_m and the relative concentration of the individual metabolites in blood and CSF are such that it is very likely that citrate is the primary (if not the only) *in vivo* substrate – at least for human SLC13A5 (there are several reviews on the topic). Furthermore, patients with loss-of-function mutations of SLC13A5 have a 3-4-fold increase in citrate levels in both blood and CSF. However, they do not have a similar increase in the levels of the other monocarboxylates that are presumed to be SLC13A5 substrates. The same patients have defective excretion of citrate (but not other monocarboxylates) in the urine. These data clearly point to citrate as the major (if not the only) *in vivo* substrate for human SLC13A5.

In vivo, SLC13A5 seems to work under conditions of “sub-saturation”. Therefore, the question of whether other substrates might be partially implicated in the phenotype of SLC13A5 sTg mice (a model of genetic overexpression) is a valid one. However, the fact that the inhibition of ER acetylation can significantly rescue the mouse phenotype (Figure 7) makes it very unlikely.

4. *Many foods contain citrate, whether SLC13A5 sTg mice's phenotype can be rescued or get worse by citrate supplement ?*

Response: We do not know.

5. *It's better to add the position of marker when western blotting is full film exposure, such as Figure 7D.*

Response: Done as requested.

6. What will happen when both *SLC13A5* and *SLC25A1* are overexpressed in mice ?

Response: We do not know. We have not attempted generating double transgenic mice.

7. Sometimes β -Actin might have two strips, more often there is only one strip, such as Figure 7B (*SLC25A1* sTg) and Figure 7C. Why are there two strips in Figure 7B (*SLC13A5* sTg) ? Are there always two strips in this kind of tissue ?

Response: More than ten isoforms of beta-actin have been reported. Four of them result in protein bands that run quite close to each other (375aa, 334aa, 332aa, 309aa). Sometimes, the antibody used in this study picks up more than one band (see the original uncropped gels in the Supplementary Material). This is mentioned by the manufacturer and is also evident from other papers.

8. Loss of *SLC13A5* has been reported to be associated with epilepsy, whether *SLC13A5* overexpression can improve related phenotypes or indicators ? Can you evaluate the potential of *SLC13A5* as a potential prescription and the potential side effect of premature aging ?

Response: We never proposed (and we do not propose) *SLC13A5* as a target for human aging. In humans, both loss-of-function mutations and gene duplication events affecting *SLC13A5* are associated with severe (but different) disease phenotypes. These two situations should be targeted differently. Furthermore, progeria is a disease, and we should always be careful in translating “progeria data” into “normal aging data”.

9. In general, increased lipid accumulation in the liver tends to mean heavier liver weight. Can you explain why *SLC25A1* sTg mice have more lipid droplets, but less weight of liver ? Could this be caused by a loss of total body weight of *SLC25A1* sTg mice ?

Response: We do not know, but it is a valid possibility.

10. *SLC13A5* sTg mice display a progeria-like phenotype, in this paper you have used some experiments to describe it, for *SLC25A1*, why did its overexpression reduce the size of mice so much in Figure 1C ? What kind of disease might be involved ?

Response: We do not know the cause of the reduced body size of *SLC25A1* sTg mice. We hope to obtain more information with follow-up studies.

Reviewer #3

1. Why did the authors choose to measure acetylation stoichiometry instead of relative change of acetylation abundance? Acetylation is widely recognized as an extremely low stoichiometric event, as most acetylation occurs at very low stoichiometry (median 0.02%), high stoichiometry acetylation (>1%) only occurs on nuclear proteins involved in gene transcription and on

acetyltransferases (PMID: 30837475). The regular measurement error of MS intensity is already much larger than 1%, making the accuracy of stoichiometry measurements, particularly after subcellular fractionation, likely to be very poor. For example, a 1% acetylation stoichiometry can be easily measured and calculated as 2% (or even larger) in MS quantification, which can generate an artificial 2-fold change in stoichiometry calculation. This is supported by the data presented in Figure 4B and 4C, which show patterns for random changes in stoichiometry rather than the expected increase in transgenic mice. In addition to the fold-change, the original stoichiometric values need to be displayed.

Response: Using our stoichiometry methods, we respectfully point out our published observations (PMID: 36199173, PMID: 35203088, PMID: 35146426, PMID: 33846551, PMID: 33479349, PMID: 32290654, PMID: 31477734, PMID: 24917678, PMID: 25555129). As we have discussed in our previous work (PMID: 32290654), all current MS stoichiometry measurements suffer some caveats. Antibody enrichment can reflect antibody bias and magnify large fold changes in protein sites with extremely small stoichiometries. Our labeling method and analysis are most accurate for sites above 1% stoichiometry. In general, we are most interested in sites that show significant stoichiometry change above 1%. While we do document all acetylation sites with measurable stoichiometry (see uploaded datasets), we do not use these to interpret the biology. Yes, most lysine residues are not significantly acetylated, but that is entirely expected for a regulatory modification. We choose to focus on the sites with significant stoichiometry (>1%) and that demonstrate a statistically significant change under a perturbed condition.

The reviewer states that Figure 4 data “*show patterns for random changes in stoichiometry rather than the expected increase in transgenic mice.*” As we have demonstrated previously, protein acetylation dynamics are complex, and under conditions where acetylation is dramatically increased on certain proteins and sites, there are other protein sites that show a simultaneous decrease (PMID: 36199173, PMID: 35203088, PMID: 35146426, PMID: 33846551, PMID: 33479349, PMID: 32290654, PMID: 31477734, PMID: 24917678, PMID: 25555129). Thus, a rise or drop in acetyl-CoA does not equate to all acetylation sites reflecting the exact same behavior. These changes are biologically significant and cannot be defined as “*random*”.

The original stoichiometric values are displayed in our uploaded datasets.

The following paragraph has been added in the Discussion section:

*...As mentioned above, fluctuations in acetyl-CoA availability occurring within the range of affinity constants of different Nε-lysine acetyltransferases enable substrate-mediated regulation of protein acetylation^{1,3,5}. As a result, changes in acetyl-CoA levels are not immediately mirrored by similar changes in lysine acetylation of every protein and site. In other words, under conditions where acetylation is dramatically increased on certain proteins and sites, there are other proteins and sites that show a simultaneous decrease^{29,30,34,49,50}. This behavior is clearly manifested at the level of the acetyl-proteome where, within the same mouse model and tissue, we observed both increased and decreased stoichiometry of acetylation (see **Figure 4**, **Figure 5**, **Figure S2**, and **Figure S4**). This differential response occurred in the face of a general trend toward increased acetylation (see **Figure 4**, **Figure S2**, and **Figure S4**). In other words, the cell can direct the adaptive response to the increased availability of citrate and acetyl-CoA towards certain biological and biochemical pathways...*

2. It is important to evaluate the reaction efficiency of chemical acetylation using acetic anhydride. Specifically, it is important to determine whether all free lysine residues were acetylated by the isotope-labeled reagent in the proteomics data. If not, it is crucial to calculate the labeling efficiency, as incomplete labeling could lead to significant errors in stoichiometry calculations.

Response: The reviewer is absolutely correct in stating that the chemical efficiency of acetic anhydride labeling is important for the quantification. That is why we follow our established methods as discussed in (PMID: 24917678, PMID: 32290654), which documents how we ensure ~100% labeling.

3. It is recommended that the authors examine the total acetylation levels of different tissues in both wild-type and transgenic mice using a pan-anti-acetyl-lys antibody.

Response: We have added Figure S8 showing increased levels of lysine acetylation in the nucleus but overall normal levels of lysine acetylation in total liver extract. These results provide further support to our conclusion that “...the cell can direct the adaptive response to the increased availability of citrate and acetyl-CoA towards certain biological and biochemical pathways...” The data in Figure S8 and Figure 7D were obtained with the same pan-anti-acetyl-lys antibody and from the same tissue (liver) to ensure experimental consistency. The results are clearly in line with the acetylotomic data displayed in Figure 4, Figure 5, and Supplementary Figure 3.

The following two paragraphs have been added in the Results section:

...SLC13A5 sTg mice also displayed increased N ϵ -lysine acetylation of nuclear-associated proteins but normal levels of global acetylation within the total lysate (**Figure S8**). Interestingly, the increased acetylation of the nuclear compartment was not as dramatic as the ER compartment (compare **Figure 7D** to **Figure S8A**). Overall, these results point, again, to a selective adaptive response and appear to reflect the acetyl-proteome data (**Figure 4 and 5**)...

...As expected, C9 did not rescue the hyperacetylation in the nucleus thus confirming its specificity toward the ER-based ATases (**Figure S8A**)...

The following two paragraphs have been added in the Discussion section:

...As mentioned above, fluctuations in acetyl-CoA availability occurring within the range of affinity constants of different N ϵ -lysine acetyltransferases enable substrate-mediated regulation of protein acetylation^{1,3,5}. As a result, changes in acetyl-CoA levels are not immediately mirrored by similar changes in lysine acetylation of every protein and site. In other words, under conditions where acetylation is dramatically increased on certain proteins and sites, there are other proteins and sites that show a simultaneous decrease^{29,30,34,49,50}. This behavior is clearly manifested at the level of the acetyl-proteome where, within the same mouse model and tissue, we observed both increased and decreased stoichiometry of acetylation (see **Figure 4, Figure 5, Figure S2, and Figure S4**). This differential response occurred in the face of a general trend toward increased acetylation (see **Figure 4, Figure S2, and Figure S4**). In other words, the cell can direct the adaptive response to the increased availability of citrate and acetyl-CoA towards certain biological and biochemical pathways...

...As with the stoichiometry of acetylation, we did not observe a generalized increase in lipid levels but rather a divergent response at the class, sub-class, and specie level. This divergent response was also very different across the tissues examined (see **Figure 6** and **Figures S5-S7**)...

4. The performance of subcellular fractionation should be assessed by examining subcellular-specific protein markers.

Response: We wish to point out that we performed an ER enrichment, which is different from the classical subcellular fractionation. ER enrichment is a well-established protocol and was performed with a commercially available kit (see the Materials section). We have performed this enrichment many times (as documented in PMID: PMC6156544, PMID: PMC6718414, PMID: PMC8041774, PMID: PMC8042170, PMID: PMC8881600) and we have always obtained consistent results. We do not typically publish the markers. However, a typical calnexin (ER marker) staining following enrichment is shown below for reference.

Reviewers' comments:

Reviewer #1 (Remarks to the Author):

As far as I am concerned, the manuscript is now acceptable to be published.

Reviewer #2 (Remarks to the Author):

SLC13A5 and SLC25A1 are two important citrate transporters. Many studies have reported pathological conditions caused by deletion of these two genes. This paper systematically describes the phenotype of mice with systemic overexpression of SLC13A5 and SLC25A1, just as Table 1 is listed. The phenotypes of SLC13A5 and SLC25A1 sTg mice are very strong and significant, there are many phenotypes that deserve further research. In this paper, Nε-lysine acetylation profiles and lipid profiles of two types of mice were compared using omics methods, it reveals that different sources of citrate may have different metabolic responses, this also illustrates the complexity of citrate metabolism. I have read the responses to the review comments. The author answered question 2,3,7 and 8 in detail and had a complete discussion. As for question 1,4,6,9 and 10, the author did not give a definite answer. I understand the difficulty in answering those questions, and that could be another story. I also noticed the replies to other reviewers, the author cited a lot of literature methods and gave detailed explanations.

This paper may draw more attention to the Slc13a5 and Slc25a1 genes, and many phenotypes still need further study. There are still a lot of work to be done around these two genes. I think this article makes sense, at least in terms of the phenotypic findings.

Reviewer #3 (Remarks to the Author):

This reviewer would like to thank the authors for providing explanations and additional data to improve the clarity of their manuscript. However, there are still concerns regarding the data on acetylation stoichiometry.

First, I remain unconvinced that the acetylomic data presented in Figure 4c aligns with other observations reported by the authors, such as increased ac-CoA levels and increased Kac levels in specific organelles. The authors mention that "The original stoichiometric values are displayed in our uploaded datasets." However, the datasets are missing in the revised manuscript. Furthermore, it is crucial for the authors to present stoichiometric values for all replicates along with their respective standard deviations. Additionally, it would be helpful to calculate the coefficient of variation (CV) values within each group and across all groups. It is important to note that the CV of inter-group should be significantly larger than the intra-group CVs to ensure a reliable dataset for biological changes.

Regarding the previous question (Q2), please provide the calculated labeling efficiency of this study instead of simply citing a previous publication.

I do not have a password to access the raw data deposited by the authors at ProteomeXchange Consortium (PXD040008).

Point-by-Point Response

We wish to thank the Editor and the Reviewers for their contribution to our manuscript. Changes within the manuscript are highlighted. A comprehensive point-by-point response to the suggestions and questions can be found below.

Reviewer #1

We thank Reviewer #1 for the positive evaluation of this manuscript.

Reviewer #2

We thank Reviewer #2 for the positive evaluation of this manuscript.

Reviewer #3

The authors mention that "The original stoichiometric values are displayed in our uploaded datasets." However, the datasets are missing in the revised manuscript.

Response: The datasets included in this manuscript have already been deposited and will become freely accessible after publication. Credentials to access them were provided during the submission of both the original and the revised version of the manuscript. We apologize if the Reviewer was not able to retrieve them from the submitted material. To facilitate their retrieval, the full credentials are also included below:

Acetylproteomics:

Username: MSV000091235_reviewer

Password: SLC_acetylomics

Lipidomics:

Username: MSV000090972_reviewer

Password: reviewer_pw

Furthermore, it is crucial for the authors to present stoichiometric values for all replicates along with their respective standard deviations.

Response: They are already included in the full datasets. See above to access them.

Additionally, it would be helpful to calculate the coefficient of variation (CV) values within each group and across all groups. It is important to note that the CV of inter-group should be significantly larger than the intra-group CVs to ensure a reliable dataset for biological changes.

Response: CV values are included in the datasets.

Regarding the previous question (Q2), please provide the calculated labeling efficiency of this study instead of simply citing a previous publication.

Response: We always perform two rounds of labeling with the heavy labeled acetic anhydride, with a pH adjustment between to maximize the labeling efficiency of acetic anhydride. Under these conditions, we obtain a >99% labeling efficiency. To address the Reviewer's concern, we have reanalyzed the data to look for peptides where the lysine residue was unlabeled, and trypsin was able to cleave. As expected, these peptides accounted for less than 1% of the total

peptide signal, thus confirming that indeed we have a >99% percent labeling efficiency. We have added language to the Methods Section to indicate the >99% labeling efficiency (see highlighted sentence in the revised manuscript).

I do not have a password to access the raw data deposited by the authors at ProteomeXchange Consortium (PXD040008).

Response: See above.